

# Multi-scale soil moisture data and process-based modeling reveal the importance of lateral groundwater flow in a subarctic catchment

Jari-Pekka Nousu[1,2], Kersti Leppä[2], Hannu Marttila[1], Pertti Ala-aho[1], Giulia Mazzotti[3],
Terhikki Manninen[4], Mika Korkiakoski[5], Mika Aurela[5], Annalea Lohila[5,6], and Samuli Launiainen[2]

[1]Water, Energy and Environmental Engineering Research Unit, P.O. Box 4300, 90014 University of Oulu, Finland
[2]Bioeconomy and Environment, Natural Resources Institute Finland, Helsinki, Finland
[3]Univ. Grenoble Alpes, Université de Toulouse, Météo-France, CNRS, CNRM, Centre d'Études de la Neige, Grenoble,
France
[4]Meteorological Research, Finnish Meteorological Institute, P.O. Box 503, 00101, Helsinki, Finland
[5]Climate System Research, Finnish Meteorological Institute, P.O. Box 503, 00101, Helsinki, Finland
[6]Institute for Atmospheric and Earth System Research INAR, University of Helsinki, Helsinki, Finland

**Correspondence:** Jari-Pekka Nousu (jari-pekka.nousu@luke.fi)

**Abstract.** Soil moisture plays a key role in soil nutrient and carbon cycling, plant productivity and in energy, water, and greenhouse gas exchanges between the land and the atmosphere. In this study, we used the Spatial Forest Hydrology (SpaFHy) model, *in-situ* soil moisture measurements and Sentinel-1 SAR-based soil moisture estimates to explore spatiotemporal controls of soil moisture in a subarctic headwater catchment in northwestern Finland. The role of groundwater dynamics and lateral

flow on soil moisture was studied through three groundwater model conceptualizations: i) omission of groundwater storage and lateral flow, ii) conceptual TOPMODEL approach based on topographic wetness index, and iii) explicit 2D lateral groundwater flow. The model simulations were compared against continuous point-scale measurements, distributed manual measurements conducted in the study area, and novel SAR-based soil moisture estimates available from the area at high spatial and temporal resolution. Based on model scenarios and model-data comparisons, we assessed when and where the lateral groundwater flow

shapes soil moisture, and under which conditions soil moisture variability is driven more by local ecohydrological processes, i.e. the balance of infiltration, drainage and evapotranspiration. The choice of groundwater conceptualization was shown to have a strong impact on the modeled soil moisture dynamics within the catchment. All model conceptualizations captured the observed soil moisture dynamics in the upland forests, but accounting for the lateral groundwater flow was necessary to reproduce the saturated conditions commonly occurring on the peatlands and occasionally on lowland forest grid-cells.

We further highlight the potential of integrating multi-scale observations, including spatially explicit remote sensing data, with land surface and hydrological models. The results have broad implications for choosing suitable models for studying ecohydrological and biogeochemical processes as well as earth system feedbacks in subarctic and boreal environments.

## 1 Introduction

Soil moisture has a direct influence on land surface energy fluxes (Seneviratne et al., 2010; Ji et al., 2017), partitioning of

precipitation into infiltration and runoff (Liu et al., 2019; Singh et al., 2021), and plant productivity and water use (Daly and





Porporato, 2005; Lagergren and Lindroth, 2002). It is also a key variable controlling soil microbial activity and consequent greenhouse gas emissions (Bonan, 1990; Karhu et al., 2014; Lohila et al., 2016; Makhnykina et al., 2020), and soil carbon balance (Larson et al., 2023). In the boreal and subarctic region, the climate change is predicted to amplify seasonal variability of soil moisture due to longer and more frequent summer droughts, increased autumn and winter precipitation (Holmberg

et al., 2014; Ruosteenoja et al., 2018), and changes in snow accumulation and melt (Räisänen, 2021). The altered soil moisture dynamics have an effect on the severity of abiotic stressors (e.g. water shortage, excess water, extreme temperatures) and biotic damages, both affecting tree health, mortality and forest productivity (Buermann et al., 2014; Muukkonen et al., 2015; Wang et al., 2023). The changes in soil moisture across the landscape can significantly impact vegetation dynamics and alter the competition between species, shaping the structures of the ecosystem (Venäläinen et al., 2020; Junttila et al., 2022; Am-

eray et al., 2023). Moreover, northern peatlands are sources of methane (Huttunen et al., 2003; Schneider et al., 2016) and boreal upland forests can turn from methane sinks to sources under long-lasting high soil moisture conditions (Korkiakoski et al., 2022; Lohila et al., 2016). Hence, accurate information on spatiotemporal soil moisture conditions has the potential to improve estimates of tree health, carbon and greenhouse gas sinks and sources, and nutrient leaching (Bond-Lamberty et al., 2016; Nakhavali et al., 2021). Soil moisture dynamics is also critical for weather and hydrological forecasting (Zhang et al.,

2020a; Joo and Tian, 2021), climate change impact studies (Seneviratne et al., 2010; Kløve et al., 2014; IPCC, 2019), and for developing sustainable forest management practices (Salmivaara et al., 2021; Kankare et al., 2019).

Soil moisture has strong spatiotemporal variability driven by hydrometeorological conditions, landscape heterogeneity, and hydrological connectivity through lateral groundwater flow (Corradini, 2014; Kemppinen et al., 2023; Kim and Mohanty, 2016; Ji et al., 2017). The unsaturated soil is bounded at the bottom by the water table, and exchanges between the saturated

and unsaturated zone occur through upward capillary rise and downward percolation (Maxwell et al., 2007; Miguez-Macho et al., 2007; J.-P. Vergnes and Habets, 2014). The lateral groundwater flow and consequent undulation of the water table influences soil moisture especially in areas with shallow water table such as riparian areas, floodplains, and peatlands (Krinner, 2003; Decharme et al., 2019; Kollet and Maxwell, 2008).

Information on soil moisture dynamics can be obtained via *in-situ* measurements and remote sensing, as well as using

numerical models (Robinson et al., 2008; Yu et al., 2021; Dobriyal et al., 2012). Continuous automatic *in-situ* measurements are well suited to capture soil moisture patterns at high temporal resolution at point-scale (Moreno et al., 2022; Kemppinen et al., 2023). However, distributing the observation network in space requires significant resources (Tyystjärvi et al., 2022) and is thus restricted to specific study areas (Kemppinen et al., 2023). Recent advances in satellite remote sensing have shown the potential to obtain soil moisture estimates at high spatial resolution (e.g. Sentinel-1 Synthetic Aperture Radar (SAR)), but their

accuracy for high-latitude forests is still limited (Celik et al., 2022). Indeed, observational methods are proven to be useful to interpret the present and historical states but cannot inform future trends and conditions. To predict soil moisture conditions under environmental change, process-based hydrological models are a prerequisite. However, their development also relies largely on observations (Panday and Huyakorn, 2004; Tyystjärvi et al., 2022), and it is widely accepted that the integration of *in-situ* measurements, remote sensing, and process-based modeling is the best avenue forward (Crow and Yilmaz, 2014; Sidle,





2021). To yield accurate predictions, it is essential that process-based models represent the most relevant local features and processes that affect soil moisture dynamics (Sidle, 2021; Ji et al., 2017; Kollet and Maxwell, 2008).

Due to the proliferation of geospatial data on land use, topography, vegetation, and soil characteristics, spatially distributed models can, to an increasing extent, incorporate spatial variability in their parameterizations, and allow extending point-scale simulations to scales relevant for practical applications (Launiainen et al., 2019; Ma et al., 2016; Clark et al., 2015; Maneta

and Silverman, 2013). To model soil moisture at high spatial resolution, incorporating the effects of local soil texture and vegetation, as well as the conceptualization of subsurface water storage and lateral flow become important. Integrated surface-groundwater models can explicitly represent these interactions in 3D (Ala-aho et al., 2017a; Thornton et al., 2022; Autio et al., 2023), but are rarely used in ecosystem studies or large-scale applications due to their vast data needs and low computational efficiency. For instance, attention towards groundwater dynamics is rather recent in land surface models used in climate,

weather and hydrological modeling communities (Decharme et al., 2019; Kollet and Maxwell, 2008; Maxwell and Condon, 2016; Ji et al., 2017; Niu et al., 2014). In catchment hydrological models, the lateral movement of groundwater is still rarely explicitly addressed, and the groundwater dynamics are often based on simplified conceptual approaches such as the use of topographic wetness index (TWI) (Beven and Kirkby, 1979) or grid-cell independent groundwater buckets (Bergström, 1992). These simplified approaches can efficiently link grid-cell and catchment water budgets and simulate sufficient discharge

dynamics (Launiainen et al., 2019). They can also accurately estimate soil moisture dynamics in landscape locations, where water balance is mostly driven by local processes, i.e. infiltration, vertical water percolation and evapotranspiration (ET), rather than lateral flow and capillary rise (Tyystjärvi et al., 2022). However, once the impacts of lateral groundwater flow and a shallow water table become more pronounced, models neglecting lateral flow encounter obvious challenges to accurately simulate soil moisture dynamics (Kollet and Maxwell, 2008). Consequently, they often exhibit dry biases that directly affect simulations of

soil evaporation and plant transpiration (Maxwell and Condon, 2016).

Hydrological models are advancing towards incorporating more processes at higher spatial resolution (Sidle, 2021; Wood et al., 2011), but model calibration and evaluation are still largely based on point-scale observations of soil moisture, ET and stream discharge at catchment outlet (Ala-aho et al., 2017b; Launiainen et al., 2019), creating uncertainties for spatiotemporal simulations (Koch et al., 2018). The persistent lack of spatial observations of hydrological fluxes and water storages (model

state variables) prevents leveraging the full potential of distributed models and available data on landscape characteristics. Recent advances in remote sensing to produce spatially explicit data of precipitation (Yu et al., 2022) and ET (Bhattarai and Wagle, 2021), canopy and soil water content (Manninen et al., 2021; Zhang and Zhou, 2015), snow properties (Meriö et al., 2023), and water table depth (Toca et al., 2023; Räsänen et al., 2022; Isoaho et al., 2023) open new opportunities to evaluate (Niu et al., 2021) and in some cases calibrate (Koch et al., 2018) spatially distributed models. Such data are increasingly

included in hydrological model-data assimilation (Li et al., 2023; Deschamps-Berger et al., 2022).

In this study, we assess the controls of spatiotemporal soil moisture at the subarctic Pallas Lompolojängänoja headwater catchment in northern Finland. We combine analysis of multi-scale observations, including *in-situ* continuous and manual soil moisture measurements (Marttila et al., 2021; Aurela et al., 2015), SAR-based spatiotemporal estimates (Manninen et al., 2021), and process-based hydrological modeling. We use the Spatial Forest Hydrology model (SpaFHy; Launiainen et al.,





2019) at high spatial resolution ($16 \times 16$ m$^2$) with three alternative conceptualizations for groundwater storage and dynamics. In particular, we focus on the influence of a shallow water table and lateral groundwater flow, as well as vegetation heterogeneity on spatiotemporal soil moisture dynamics through the following research questions:

1. Where and when does lateral groundwater flow affect the temporal variability of top soil moisture?

2. How does the role of lateral groundwater flow compare to the impact of vegetation heterogeneity in shaping soil moisture
patterns?

3. How do SAR-based soil moisture estimates compare with the models, and can they be useful in model evaluation?

To answer the research questions, we compare and contrast model predictions i) between model conceptualisations, ii) against point-scale measurements of soil moisture, and iii) against SAR-based soil moisture estimates available from the study area.

## 2 Materials and methods

### 2.1 Study site

Our study area is located in the Pallas area (67°59'N 24°13'E) in northwestern Finland (Fig. 1B,C). Pallas has over 85 years of meteorological observations (Lohila et al., 2015), and the area has been recently set up as an interdisciplinary platform for atmospheric, ecological, and hydrological research. It includes multiple eddy covariance (EC) stations measuring surface-atmosphere energy and greenhouse gas fluxes, and both manual and automated ecohydrological monitoring at range of ecosystem types (Marttila et al., 2021). The climate in the area is characterized as subarctic. The long-term annual (1991–2020) mean
temperature and mean annual precipitation at the Muonio weather station, located approximately 25 km west of Pallas, are -0.6 °C and 532 mm, respectively (Jokinen et al., 2021). A seasonal snow cover persists from about October until May (Aurela et al., 2015). Particularly, we consider the Lompolojängänoja catchment (hereafter LJO, Fig. 1A), which has the total area of circa 4.5 km$^2$ with altitudes varying between 268 m and 364 m a.s.l. Soils in the upland parts of the catchment are mainly
gravely sand and sandy tills, and vegetation cover varies from coniferous forests to various types of mires such as open fens, treed mires, and paludified forests. The area has had little human influence and can be considered a pristine subarctic headwater catchment. Figure 1A gives an overview of the landscape and the main measurement locations in the LJO catchment.

   EC flux data and the meteorological data used in this paper was collected from two Finnish Meteorological Institute (FMI) flux stations located in the catchment of Lompolojängänoja. The forest site, Kenttärova (ICOS Ecosystem associate site), is a
Norway spruce-dominated forest growing on podzol soil with the age of the trees varying from 90 to 250 years. The number of trees, 643 and 68 live stems/ha for spruce and deciduous trees (mainly Betula pubescens), has stayed the same since a survey in 2011 (Aurela et al., 2015). The dominant tree height is currently about 15.5 m and 11 m for spruce and deciduous trees, respectively. In 2011, a mean one-sided leaf-area index (LAI) for Norway spruce and birch was 2.0 and 0.1 m$^2$ m$^{-2}$, respectively (Aurela et al., 2015).





The mire site, Lompolojänkkä (ICOS Ecosystem Class 2 site) is an open, mesotrophic sedge fen (Zhang et al., 2020b) with a maximum peat thickness of about 2.5 m (Mathijssen et al., 2014). The Lompolonjängänoja stream flows through the long and narrow fen, draining into a nearby lake Pallasjärvi. According to a plant species and coverage measurement done at 200 points around the EC tower in 2018, the dominant vascular species growing at the site and having a mean coverage higher than 0.5 % were Andromeda polifolia, Betula nana and B. pubescens, Carex spp., Equisetum spp., Eriophorum spp. The dominant

moss species are Sphagnum spp., whose coverage is about 50 %. In 2018 sampling, the mean one-sided LAI and the vegetation height were on average 1.4 $m^2$ $m^{-2}$ and 40 cm, respectively.

## 2.2   Models

We used the Spatial Forest Hydrology model (SpaFHy; Launiainen et al., 2019), developed to predict spatial and temporal patterns of hydrological fluxes and state variables in the vegetation canopy, organic moss-humus layer, and rootzone (top soil).

Its original aim was to provide a simple and practically applicable framework to study the effects of landscape heterogeneity and management on catchment hydrology in boreal forests. SpaFHy was tested for 9 EC-flux sites in Finland and Sweden (stand-scale; ET and soil moisture) and for 21 small boreal headwater catchments in Finland (catchment-scale; runoff dynamics and ET to precipitation ratio) in Launiainen et al. (2019). Lately, it has been adapted to drained peatland forests (Leppä et al., 2020; Stenberg et al., 2022), extended with nutrient balance and leaching modules (Laurén et al., 2021), applied to model forest

drought risks (Launiainen et al., 2022) and used to predict soil moisture dynamics in the arctic tundra (Tyystjärvi et al., 2022).

The original SpaFHy includes two groundwater conceptualizations: a free drainage approach (i.e. neglecting groundwater dynamics, SpaFHy-1D) and a TOPMODEL-based approach (i.e. groundwater return flow based on topographic wetness index, SpaFHy-TOP). In this study, we implemented a new submodel to represent the 2D lateral groundwater Darcy flow (SpaFHy-2D). The salient features of the three model versions are briefly described next and summarized in Table 1. The general model

parameters are given in Table 2.

### 2.2.1   SpaFHy-1D

SpaFHy-1D considers grid-cells as independent hydrological units (Launiainen et al., 2019). The hydrological processes in the vegetation canopy, snowpack, organic moss-humus layer, and rootzone are explicitly simulated at a daily timestep for each grid-cell in the model domain. The above-ground fluxes and state variables are computed in the canopy submodel, including rainfall

and snowfall interception and evaporation, throughfall, transpiration, and snow accumulation and melt. The bucket submodel describes soil hydrology and soil moisture dynamics in two layers. The upper layer is the organic moss-humus layer, whose water budget is affected by interception and evaporation, as well as infiltration to the lower rootzone layer, where drainage and transpiration take place. The lateral water flow between the grid-cells is omitted, and drainage from the bucket submodel is removed from the model domain as stream discharge at the catchment outlet. Thus, SpaFHy-1D represents a situation

where soil moisture variability within the catchment is driven only by the heterogeneity of vegetation, soil characteristics, and meteorological forcing. Similar conceptualizations of soil hydrology are common for large-scale land surface models (Smith et al., 2001; Noilhan and Mahfouf, 1996).



**Figure 1.** The Lompolojängänoja (LJO) catchment and its hydrological measurements (A, the aerial image by NLSF, 2020) is located in the northern boreal zone (B, green area, Olson et al., 2001) in northwestern Finland (C, Esri, 2023). The ICOS flux stations Kenttärova and Lompolonjänkkä are presented in red circles, stream gauge in blue, soil moisture measurement locations are labeled and presented in light blue, and water table depth monitoring locations are labeled and presented in orange.





### 2.2.2 SpaFHy-TOP

SpaFHy-TOP includes a conceptual description of the saturated zone using the TOPMODEL approach (Beven and Kirkby,
1979). Drainage from the bucket submodel feeds TOPMODEL's lumped catchment groundwater storage, which is then spatially distributed via the topographic wetness index (TWI). The TWI is defined as the natural logarithm of the flow accumulation area (i.e. upslope area draining through the grid-cell) divided by the tangent of the local slope. Moreover, the local saturation deficit is related to the TWI and catchment average saturation deficit, creating a higher probability for grid-cells with greater TWI to become saturated. During a model timestep, return flow from the groundwater storage to the rootzone and organic
moss-humus layer occurs in grid-cells where the local saturation deficit is zero (Launiainen et al., 2019). The return flow is routed through the rootzone and the organic moss-humus layer and their respective soil moisture is updated, while the potential excess water becomes surface runoff. Catchment discharge is the sum of the catchment average baseflow (predicted by TOPMODEL) and surface runoff. This version of SpaFHy-TOP is identical to the one used in Launiainen et al. (2019, 2022).

### 2.2.3 SpaFHy-2D

SpaFHy-2D version was developed in this study to include an explicit description of the lateral groundwater flow within the catchment. The modeling domain consists of soil columns whose relative elevation to one another is defined by the digital elevation model. Each soil column extends to an impermeable layer (no-flow boundary) at a predefined depth, while the columns are characterized by their water retention characteristics (following the van Genuchten -model; van Genuchten, 1980) and saturated hydraulic conductivity based on the soil type.

Lateral flow in the saturated zone is solved using the 2D groundwater flow equation:

$$C\frac{\partial h}{\partial t} = \frac{\partial h}{\partial x}\left(T\frac{\partial h}{\partial x}\right) + \frac{\partial h}{\partial y}\left(T\frac{\partial h}{\partial y}\right) + S \tag{1}$$

where $t$ is time (d), $x$ and $y$ are the horizontal dimensions (m), $C$ is the storage coefficient (m m$^{-1}$), $T$ is transmissivity (m$^2$ d$^{-1}$), $h$ is the hydraulic head (m), and $S$ (m d$^{-1}$) is water drained from the overlaying bucket submodel. Lateral flow takes place only in the saturated zone, and thus $T$ is obtained by integrating the saturated hydraulic conductivity over the saturated layer
depth. $C$ describes the change in $h$ relative to a change in the soil column water content $W$ (m). The relation between $h$ and $W$ is solved based on the assumption that in the unsaturated zone the water content profile sets to hydraulic equilibrium (constant hydraulic head in vertical dimension; Skaggs, 1980). For numerical efficiency of solving Eq. 1, interpolation functions for $W(h)$, $T(h)$, and $C(h)$ were constructed prior to simulation for each soil column type (Laurén et al., 2021). When the soil column becomes oversaturated, i.e. groundwater level rises to rootzone, the excess water is routed as return flow to the bucket
submodel, similarly as in SpaFHy-TOP.

Streams (and ditches) in the catchment were described as cells with constant $h$, and the outflow to streams is computed based on the local hydraulic head gradient when the surrounding water table level is above the stream $h$. No flow from stream to soil is allowed. We do not consider temporal changes in stream water level and omit channel flow in the stream network; thus the





sum of the outflow into the stream cells and surface runoff form the runoff from the catchment. Catchment borders are defined
as no flow boundaries.

## 2.3  Model input

### 2.3.1  Geospatial data

To set up SpaFHy for the LJO catchment, we used mainly open geospatial data that is available throughout Finland. The rasters
used are presented in Fig. 2, and summarized in Table 1.

For canopy attributes and for distinguishing between forest soils and mires, we used the multi-source National Forest Inventory (mNFI; Mäkisara et al., 2016) data at 16 m horizontal resolution. This was also chosen as the model grid resolution for the
simulations, and other input rasters were aggregated accordingly, consistent with Launiainen et al. (2019). From mNFI data,
needle and leaf mass rasters were used to derive one-sided LAI of deciduous and coniferous trees. LAI values were estimated
using specific one-sided leaf areas for pine, spruce and birch (6.8, 4.7 and 12.0 m$^2$ kg$^{-1}$, respectively; Härkönen et al., 2015).
LAI estimates of shrub and grass were adopted from local multi-source remote sensing data by Räsänen et al. (2021). The
canopy fraction and prevailing site class (used for parameterizing the organic moss-humus layer) were also obtained from the
mNFI data.

The soil type affects the hydraulic properties of the rootzone and the SpaFHy-2D lateral groundwater flow module. A combined soil type raster was constructed by taking the peatland boundaries from the National Land Survey of Finland topographic
map NLSF (2020) and the remaining soil characteristics from the Geological Survey of Finland soil texture map (GSF, 2020)
similarly to Launiainen et al. (2019).

The catchment was delineated based on the digital elevation model (NLSF, 2020) with Whitebox GAT software (Lindsay,
2014). TWI was calculated using the slope and flow accumulation raster, with the flow accumulation determined through the
D8 method (O'Callaghan and Mark, 1984). The stream network was obtained from NLSF (2020). Furthermore, topographic
impacts for solar radiation were considered by computing a daily shading coefficient, calculated as the potential daily radiation
input for each grid-cell normalized by the potential input at the grid-cell of the Kenttärova station, where the global radiation
forcing was measured (see Sect. 2.3.2).

### 2.3.2  Meteorological forcing

All SpaFHy versions require the same daily meteorological forcing: mean air temperature $T_a$ (°C), global radiation $R_g$
(Wm$^{-2}$), relative humidity RH (%), wind speed U (m$^{-1}$) and daily accumulated precipitation P (mm). This data was compiled
and made available by Nousu et al. (2023), and includes *in-situ* observations at Kenttärova station available from the Finnish
Meteorological Institute (FMI) open database (FMI, 2021), supplemented by FMI's $R_g$ observations from the Kenttärova station (located at the hilltop, Fig. 1). The data gaps in $R_g$ were first filled by data from contiguous sites and then by ERA5
reanalysis data (Hersbach et al., 2020). We multiplied the $R_g$ forcing by the shading coefficient (see Sect. 2.3.1) for each day

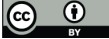

**Figure 2.** Set of geospatial rasters used to set up the model for the LJO catchment. Top row shows leaf-area index (LAI) for different plant types. For each grid-cell, the conifer and deciduous LAI forms the canopy LAI, and understory LAI is the sum of shrubs and grasses. TWI is the topographic wetness index. The rasters overlay a topographic map (NLSF, 2020).





| Geospatial data | Canopy | Bucket | TOPMODEL | 2D Flow |
|---|---|---|---|---|
| Digital elevation model | | | ✓ | ✓ |
| Catchment mask | ✓ | ✓ | ✓ | ✓ |
| Topographic wetness index | | | ✓ | |
| Shading coefficient | ✓ | ✓ | | |
| Site class | | ✓ | | |
| Leaf area index | ✓ | | | |
| Canopy height | ✓ | | | |
| Canopy fraction | ✓ | | | |
| Soil type | | ✓ | | ✓ |
| Streams | | | | ✓ |
| **Model configuration** | **Canopy** | **Bucket** | **TOPMODEL** | **2D Flow** |
| 1D | ✓ | ✓ | | |
| TOP | ✓ | ✓ | ✓ | |
| 2D | ✓ | ✓ | | ✓ |

**Table 1.** Geospatial data used by each submodel, and submodel used by each model configuration.

(Fig. 2) to account for the topographic effects on the radiation forcing at each grid cell. For the other meteorological variables, a spatially uniform forcing was applied.

## 2.4 Model parameterization

The canopy and bucket submodels were common to all model versions and parameterized as in Launiainen et al. (2019). The only exception was the organic moss-humus layer, which was refined to allow for full or partial saturation in situations where
upward return flow occurs from the rootzone layer. Drainage from the organic moss-humus layer to the rootzone layer is represented identically as the drainage from the rootzone layer (Eq. 18 in Launiainen et al. (2019)). To account for the different hydraulic properties of the organic moss-humus layer in mineral forest soils (dominated by feather mosses) and peatlands (mainly Sphagnum moss), the moss hydraulic parameters (porosity, field capacity, and relative available water parameter) were derived from Williams and Flanagan (1996) and Elumeeva et al. (2011). The peatlands and mineral soils were separated based
on mNFI site class (see Fig. 2); the site class and soil type -specific parameters for the organic moss-humus, rootzone and deep -layers are presented in the Supplement (Table S1–S3). Available literature was used to define the van Genuchten (1980) water retention parameters for the Bucket and 2D Flow submodels (Autio et al., 2023; Menberu et al., 2021). Due to a lack of reliable data on the depth-to-bedrock, a uniform thickness of 5 m was assigned for the deep soil layer of the 2D groundwater module throughout the model domain. Canopy parameters for surface conductance for evaporation from the wet forest floor ($G_f$) and





canopy storage capacity for rain ($w_{max}$), and the TOPMODEL effective soil depth parameter were obtained from Launiainen et al. (2019). No further calibration of any model parameters was conducted in this study.

Model simulations with the three different treatments of groundwater dynamics (named 1D, TOP and 2D) were run with identical meteorological forcing, geospatial inputs (Fig. 2), and canopy and bucket submodel parameterizations (Table 2). To study how spatially heterogeneous vegetation affects soil moisture, additional 1D simulation was run with site class specific

mean vegetation parameters. This experiment is referred to as $1D_{homog.canopy}$, and vegetation characteristics at each grid-cell belonging to a certain site class (Fig. 2) were set to the average of that particular site class (see Fig. S2). All simulations cover period from 2011-01-01 to 2021-09-01, of which the beginning until 2013-09-01 was considered as a model spin-up period and omitted from subsequent analysis.

## 2.5   Hydrological observations

This study benefits from the extensive hydrological monitoring of the LJO catchment (Marttila et al., 2021; Aurela et al., 2015). We further conducted several campaigns to measure spatiotemporal variability of soil moisture (i.e. volumetric water content $\theta$) during 2019–2021. In particular, biweekly manual measurements at 15 different points were conducted using WET-2 and PR2 Profile Probe sensors with an HH2 readout unit (Delta-T Devices Ltd., Cambridge, U.K.) sampling soil moisture profile at depths 0 cm, 10 cm, 20 cm and 30 cm. Additionally, we conducted two extended soil moisture measurement campaigns,

including 56 additional locations. The first (2021-06-17) represents wet condition when soil moisture was still highly impacted by the snow melt. The second (2021-09-01) was conducted in early autumn conditions after a precipitation event. Both these campaigns used the sensor ML3 ThetaProbe (Delta-T Devices Ltd., Cambridge U.K.) that measures soil moisture at 5 cm depth. For the ML3 ThetaProbe sensor, soil moisture at locations with peat soils at full saturation were assigned directly to the assumed peat porosity (0.88).

In addition, we used data from continuous soil moisture sensors distributed in close proximity to the Kenttärova flux site, covering the period from 2013 to 2021. From 2013 to 2017, soil moisture was continuously measured by four ThetaProbe type ML2x sensors at 5 cm and 20 cm depths (two each) (Aurela et al., 2015). In 2017, more sensors were installed alongside the existing ones, among which we used two sensors (Soil Scout Oy, Helsinki, Finland) at depths 5 cm and 30 cm. The continuous soil moisture measurements were averaged into daily values. To overcome the inherent uncertainties in *in-situ* measurements of

soil moisture, stemming from different devices and measurement and installation procedures (Robinson et al., 2008; Dobriyal et al., 2012; Iwata et al., 2017), we present the means and variability ranges of continuous soil moisture sensors, and address these uncertainties by averaging multiple manual probings within the area of interest within approximately a 5 m radius.

ET was measured by the eddy covariance (EC) technique at the two flux stations, Kenttärova spruce forest and Lompolo-jänkkä peatland (Fig. 1A). The EC systems consist of USA-1 (METEK) 3D sonic anemometer and closed-path LI-7000 (Li-cor,

Inc.) $CO_2/H_2O$ analysers (Aurela et al., 2015). The procedure for obtaining the ET fluxes from eddy covariance data is given in detail in Aurela et al. (2015) and in Nousu et al. (2023).

The runoff was measured by the Finnish Environment Institute with a 120-degree V-notch weir at the outlet (see stream gauge in Fig. 1A). Snow data consisted of automated snow depth observations at the Kenttärova flux station, and approximately





monthly manual snow water equivalent (SWE) measurements at Kenttärova and Lompolojänkkä flux stations (Marttila et al.,
265  2021).

## 2.6  SAR-based soil moisture estimates

We used SAR-based surface soil moisture estimates from the study area. This newly derived research data set was developed
by Manninen et al. (2021), who used Sentinel-1 Synthetic Aperture Radar (SAR) ground range detected high-resolution data
to produce high-resolution spatiotemporal soil moisture estimates of midday. The soil moisture estimates are based on the
gradient boosted trees machine learning method trained with and validated against discrete and continuous in-situ soil moisture
measurements at Pallas (Manninen et al., 2021). In particular, the gradient boosted trees method was trained with manual
surface soil moisture measurements (depth = 0 cm) and continuous soil moisture measurements at deeper soil layers that were
converted to surface conditions via linear regression in order to correspond to the penetration depth of the C-band SAR signal
in soil, which is in the range of 1–5 cm (Beale et al., 2021; Nolan and Fatland, 2003). Distinct algorithms were developed for
morning and evening flyovers, both relating estimates to instantaneous midday. Manninen et al. (2021) reported RMSEs of
0.065 $m^3m^{-3}$ and 0.088 $m^3m^{-3}$, and maximum errors of 0.341 $m^3m^{-3}$ and 0.339 $m^3m^{-3}$ for morning and evening satellite
flyover estimates, respectively. Yet, the validity of the method could be checked only at 92 different locations, for which soil
moisture data was available: altogether 678 discrete values not matching the overpass times of the SAR satellite and eight points
of continuous data. Hence, Manninen et al. (2021) state that the spatially very limited in-situ dataset did not allow to conclude
that the derived soil moisture maps would have the same soil moisture accuracy in the pixel resolution as the individual pixels
used for developing and validating the soil moisture retrieval method. The reason for the morning images to have slightly
smaller RMSE values than evening images was that several discrete soil moisture measurement points were in the distal slopes
of the evening SAR images and prone to shading.

   In this study, the original irregular grid with approximately 10 m pixel spacing was averaged into a 16 m regular grid using
subpixel area weights to be compared with the model outputs.

## 2.7  Evaluation methods

Annual periods are defined as hydrological years starting from September (e.g., 2016 = 2015-09-01 to 2016-08-31). We use
performance metrics of mean absolute error (MAE), mean bias error (MBE) and coefficient of determination ($R^2$) for model-
data comparison. Moreover, we use the Kling-Gupta Efficiency (KGE) (Gupta et al., 2009) for comparing daily runoffs between
simulations and observations. Mean differences (MD) are computed to compare different simulations (i.e. the mean difference
at each grid-cell).





| Parameter | Value | Units | Explanation | Note |
|---|---|---|---|---|
| Canopy | | | | |
| $A_{max}$ | 10 | $\mu$mol m$^{-2}$ s$^{-1}$ | maximum leaf net assimilation rate | Launiainen et al. (2019) |
| $g_{1,c}$ | 2.1 | kPa$^{0.5}$ | stomatal parameter for conifers | Launiainen et al. (2015) |
| $g_{1,d}$ | 3.5 | kPa$^{0.5}$ | stomatal parameter for deciduous | Lin et al. (2015) |
| $b$ | 50 | Wm$^{-2}$ | half-saturation PAR of light response | Launiainen et al. (2019) |
| $k_p$ | 0.6 | - | radiation attenuation coefficient | Launiainen et al. (2019) |
| $r_w$ | 0.2 | - | critical relative extractable water | Lagergren and Lindroth (2002) |
| $r_{w,min}$ | 0.02 | - | minimum relative conductance | Launiainen et al. (2019) |
| $G_f$ | 0.01 | ms$^{-1}$ | surface conductance for evaporation from wet forest floor | Launiainen et al. (2019) |
| $w_{max}$ | 1.5 | mm LAI$^{-1}$ | canopy storage capacity for rain | Launiainen et al. (2019) |
| $w_{max,snow}$ | 4.5 | mm LAI$^{-1}$ | canopy storage capacity for snow | Pomeroy et al. (1998), Essery et al. (2003) |
| $K_m$ | 2.5 | mm d$^{-1}$ | melt coefficient in open area | Kuusisto (1984) |
| $K_f$ | 0.5 | mm d$^{-1}$ | freezing coefficient | Koivusalo and Kokkonen (2002) |
| $Y_{max}$ | 18.5 | °C | phenology model parameter | Kolari et al. (2007) |
| $\tau$ | 13 | d | time constant | Kolari et al. (2007) |
| $T_{0,y}$ | -4 | °C | base temperature | Kolari et al. (2007) |
| Bucket | | | | |
| $z_{s,org}$ | 0.05 | m | organic layer depth | Launiainen et al. (2019) |
| $z_{s,root}$ | 0.3 | m | root zone depth | Kalliokoski et al. (2010) |
| TOPMODEL | | | | |
| $T_0$ | 0.001 | m s$^{-1}$ | transmissivity at saturation | Launiainen et al. (2019) |
| $m$ | 0.05 | m | effective soil depth | Launiainen et al. (2019) |
| 2D Flow | | | | |
| $z_{s,deep}$ | 5 | m | deep soil layer thickness | assigned |
| $z_{stream}$ | -0.2 | m | stream water level relative to surface elevation | assigned |

**Table 2.** Parameters used by each submodel





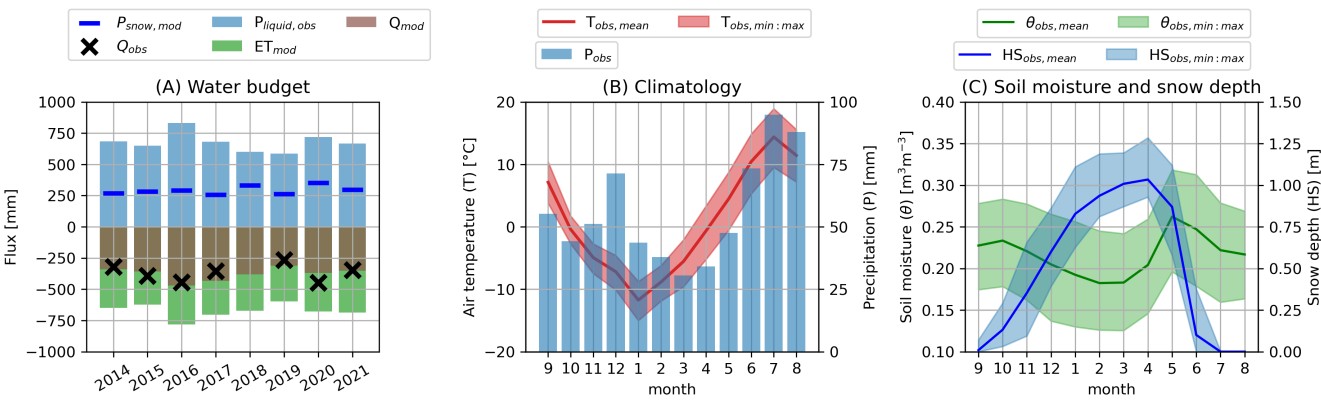

**Figure 3.** Hydrometeorological characteristics of Pallas. (A) Annual water budget as observed and simulated with SpaFHy-2D, where Q is runoff, ET evapotranspiration, P precipitation and SWE is snow water equivalent. The change in catchment water storage (including canopy water, soil water and groundwater storage) dS/dt = P + ET + Q is not shown. (B) Monthly observed climatology and (C) monthly observed volumetric soil moisture and snow depth at Kenttärova forest site. The air temperature and soil moisture envelopes represent minimum and maximum monthly averages of different years, while the snow depth envelope shows minimum and maximum of monthly maximums of different years. Due to gaps in runoff measurements in 2018, runoff observation is not presented.

## 3 Results

### 3.1 Climatology and water budget dynamics

Figure 3 introduces the main hydrometeorological characteristics of the LJO catchment. As typical for high-latitudes, the period
with permanent snow cover and freezing temperatures is long (Fig. 3B,C), with nearly half of the annual precipitation falling as snow (250 mm – 350 mm, Fig. 3A), resulting in annual peak snow depths from approximately 0.9 to 1.3 m. The snow melt period commonly spans roughly from late April to the beginning of June, resulting in the highest soil moisture values (Fig. 3C). The summer is characterized by cool to warm temperatures and higher precipitation that peaks in July (Fig. 3B).

Due to energy limitations for annual ET and large peak SWE, runoff dominates the water balance covering 49 to 67% of
annual precipitation, while ET represents 34 to 50% depending on the year (Fig. 3C). SpaFHy-2D is able to closely capture the observed annual runoff during the simulated years (Fig. 3A). Also the daily runoff dynamics are reasonably well represented by both SpaFHy-TOP (KGE: 0.63) and SpaFHy-2D (KGE: 0.65) (see Fig. S1). The summer runoff dynamics followed by precipitation events are better captured by the 2D approach whereas the baseflow is better predicted by TOP (Fig. S1). The simulations of snow water equivalent also align relatively well with the observations at Kenttärova and Lompolojänkkä (Fig.
S4). Although catchment-scale ET observations are not available, the good performance in reproducing Q/P -ratio (Fig. 3A) means annual ET is also well described. This is in accordance with the relatively good correspondence between simulated and EC-measured daily ET from Lompolojänkkä mire and Kenttärova spruce forest (Fig. S5). SpaFHy has also shown earlier to well reproduce the EC-based ET across the range of boreal and subarctic forests and peatlands (Launiainen et al., 2019).

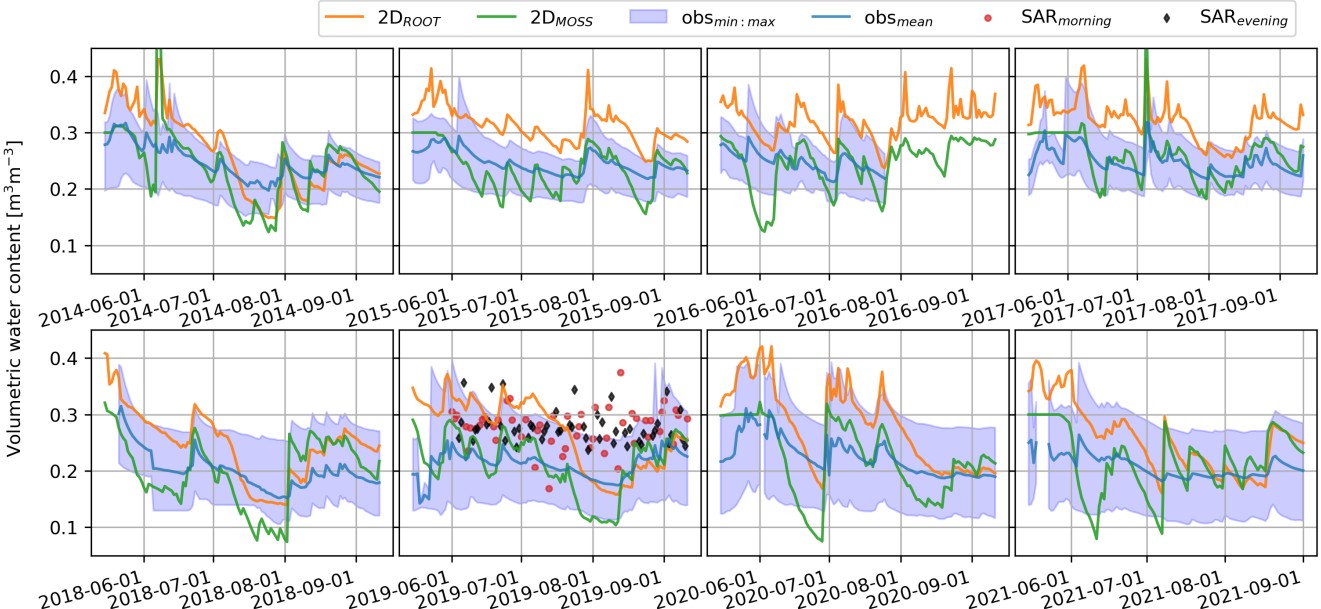

**Figure 4.** Temporal dynamics of soil moisture at Kenttärova spruce forest simulated by SpaFHy-2D (rootzone and organic moss-humus layers), measured *in-situ* and estimated from SAR for 2014–2021 during May–Sept period.

## 3.2 Comparison of temporal soil moisture dynamics at different locations

Figure 4 shows the intra-seasonal dynamics of observed and simulated (SpaFHy-2D) soil moisture at the Kenttärova site (Fig. 1A) for 2014–2021. In particular, the continuous measurements around Kenttärova are compared to the simulated and SAR-estimated mean of the nearby grid cells ($64 \times 64$ m$^2$ grid northwest of Kenttärova). The 1D and TOP predictions were nearly identical to 2D at this hilltop area that contributes to groundwater recharge but where top soil moisture is not influenced by water table dynamics. The SAR-based surface soil moisture estimates are only available in 2019. Snow melt results in high soil

moisture content in late May and beginning of June. The complete melt-out of the snowpack can be detected as the date when the modeled organic moss-humus layer moisture content begins to decrease due to evaporation. Later in the summer, the soil moisture dynamics are driven by intermittent precipitation events and more continuous drying by ET and drainage, with the general drying trend being dominant. The modeled organic moss-humus layer acts as an interception storage, and its moisture content responds quickly to precipitation and evaporation. The model overestimates the mean rootzone soil moisture content

and its temporal change compared to point-scale observations in the rootzone at Kenttärova. This mismatch could potentially be corrected by calibrating soil field capacity and wilting point. However, as the comparison only represents one grid-cell, such calibration was not considered meaningful for the aims of this study. The SAR-based soil moisture estimates mostly fall in the observed range, especially those from the morning flyovers. The SAR morning flyover is in line with the main simulated and observed temporal dynamics, drying in June and wetting in late August. However, there is noticeable noise, and the temporal





patterns of SAR morning and evening flyovers are different. A shift between the two flyovers can be noted; the wetter estimates come from the evening flyover, while the morning flyover predicts generally drier soil conditions.

     Soil moisture dynamics in 2019, with SAR-estimates available, were further assessed at eight additional biweekly measurement locations (Fig. 5). These include two forest, three peatland and three mixed forested-peatland grid-cells. The SAR-based estimates fall within the observed range corresponding to measurements at different soil depths. A systematic shift between

SAR morning and evening flyover is again noticeable (up to 0.2 $m^3m^{-3}$). The temporal variability and seasonal patterns of the SAR estimates, especially of the evening flyovers, are small and do not follow the simulated or in-situ observed moisture dynamics. The morning flyover occasionally captures some temporal dynamics observed and simulated (see e.g. Fig. 5D). The SAR-based estimates do not reach the highest observed or simulated values, likely because they integrate information from multiple signals within a given grid-cell that have been averaged to correspond to the model grid.

The SpaFHy-2D predicts the rootzone soil moisture differences between the locations reasonably well, especially in terms of ranking the locations (Fig. 5). The SAR-based soil moisture lacks the observed temporal variation whereas SpaFHy-2D simulations tend to overestimate temporal dynamics compared to the in-situ observations. The moisture content of the organic moss-humus layer above the rootzone has stronger seasonal variability, and deviates from that of soil moisture as evaporative losses exceed throughfall input leading to drying of the organic layer from mid-July to end of August (Fig. 5). SpaFHy-2D does

not include capillary rise from the rootzone layer to the organic moss-humus layer, and therefore simultaneous high evaporation and high water table can create large differences between the moisture contents of the two layers (Fig. 5D,E,G). The largest errors in rootzone simulations are in mixed forested-peatland grid-cells (Fig 5F,G) mostly due to overestimation of the water table level for the beginning of the season.

     Comparison of SpaFHy-2D simulated and in-situ observed groundwater levels is given in the Supplement (see Fig. S7).

Considering that no calibration was conducted, water table levels were simulated relatively well. Particularly, the shallow water tables are well captured but also grid-cells with deeper water table levels agree reasonably well.

### 3.3   Effect of groundwater flow conceptualizations

Scatterplots of simulated rootzone moisture content by 1D, TOP and 2D SpaFHy -versions (see Sect. 2.2) and SAR-based estimates against in-situ spatiotemporal soil moisture observations (see Sect. 2.5) across the catchment are shown in Fig. 6.

The observed soil moisture below ca. 0.55 $m^3m^{-3}$ are rather well captured by all model conceptualizations. Most of the forest grid-cells (i.e. grid-cells with high canopy fraction) belong to this category. The results indicate model performance improves when the lateral flows are accounted for, and only the 2D approach with explicit lateral groundwater flow can satisfactorily reproduce the wetter conditions above 0.55 $m^3m^{-3}$, commonly found on open peatland grid-cells, and occasionally on forest grid-cells. Conceptually, the SpaFHy-TOP should also be able to mimic groundwater dynamics via TWI. However, it is able to

capture only one of the observed wet grid-cells, and the overall goodness-of-fit metrics are close to the SpaFHy-1D version. All evaluation metrics are considerably better for the 2D model, but it tends to overestimate soil moisture on the peatland grid-cells, consistent with the overestimation in Fig 5F,G. In contrast to the models, SAR-based estimates have low correspondence with the observed soil moisture when canopy fraction is high, but match the observations better than the model on open and wetter





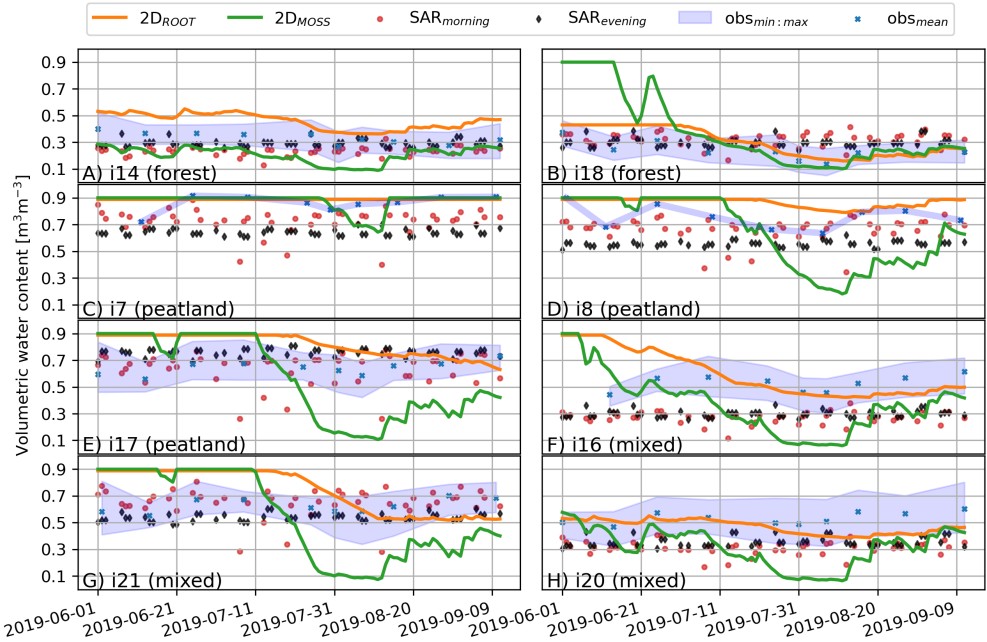

**Figure 5.** Temporal dynamics of SpaFHy-2D simulated, SAR-estimated and in-situ measured soil moisture at two forest, three peatland and three mixed forested peatland locations during June–Sept period in 2019.

grid-cells (i.e. peatlands). However, it is worth noting that the observations include all measurements in the rootzone (0–30

cm), surpassing the assumed penetration depth of the SAR signal (1–5 cm).

Qualitative spatial evaluation of the three model versions was conducted against data from the two measurement campaigns in 2021 (Fig. 7). Common to all model variants, the large-scale spatial heterogeneity of soil moisture is most strongly driven by the soil type (see Fig. 2) via the soil hydraulic properties. Particularly the differences in the 1D simulation come almost solely from differences in soil types (coarse and medium texture mineral soil and peat), while the role of vegetation heterogeneity was

minimal. The histograms of 1D simulations reveal that the moisture values are distributed around field capacities of mineral and peat soils. This is also consistent in Fig. S3 where all daily simulated distributions are shown. All model conceptualizations match rather well the drier observations in the upland forest areas, consistent with Fig. 6. However, as the 1D approach assumes independent grid-cells and neglects groundwater storage and flow, the soil moisture estimates do not reach high values as drainage rapidly removes water excessive to field capacity. Hence, the 1D simulation is biased low at wet locations (see e.g.

peatlands). Slightly more spatial variability can be seen in the TOP simulation, as it accounts for the return flow from the conceptual groundwater storage into the rootzone. Yet, the cells where return flow is activated remain rare (except close the stream network) even in wet conditions (2021-06-17) and almost non-existent in autumn conditions (2021-09-01), leading to a poor match between the model predictions and observations in wet locations.





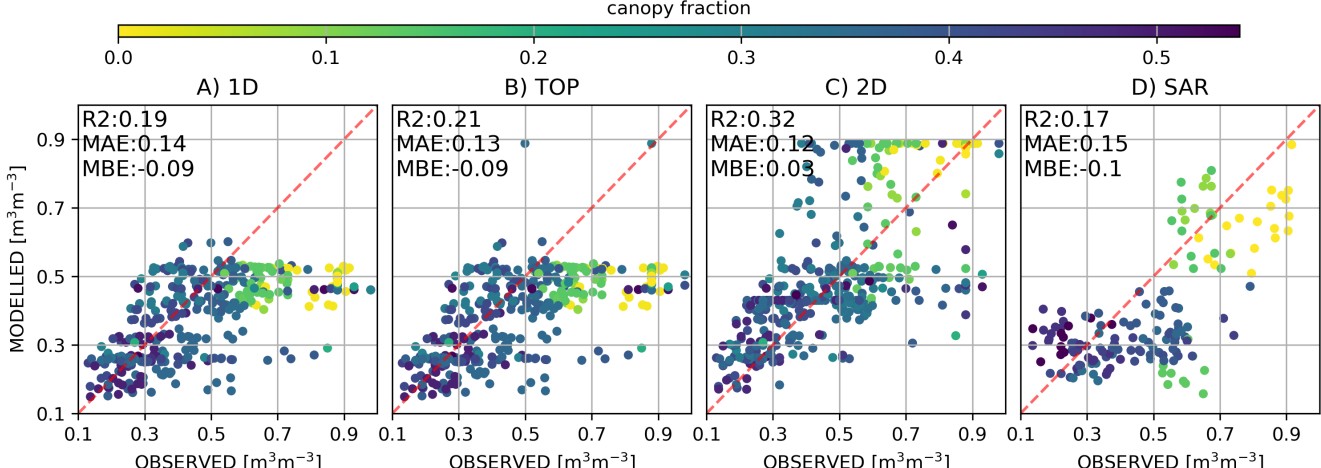

**Figure 6.** Comparison of simulated rootzone soil moisture content and SAR-based surface soil moisture estimates against spatiotemporal manual in-situ soil moisture observations. The color of the points correspond to grid-cell canopy fraction, ranging from open peatlands to forest grid-cells.

The 2D model version simulates larger saturated areas and match most of the point observations well. Nevertheless, there are still inaccuracies as saturated conditions adjacent to ditches are not well simulated, and the observed variability in the forest is not fully captured. The spatial variability of soil moisture in the 2D simulation depends strongly on the water table dynamics. Compared to other model variants, this creates stronger soil moisture variability within the catchment and yields better agreement with the observations. The histograms of Fig. 7 and Fig. S3 also show higher frequency for grid-cells to be wet.

## 3.4 Comparison of SpaFHy-2D and SAR-based estimates

Comparison against the available point-scale observations suggests that it is necessary to include the lateral groundwater flow to model the spatial patterns of soil moisture at LJO catchment. Point-scale measurements can only capture a fraction of the simulated time steps and grid-cells, and thus, a comparison with the spatially explicit SAR-based soil moisture is conducted in Fig. 8. As already noted, the simulated spatial patterns follow mostly the soil parameterizations, as well as water table dynamics affected by the lateral flow. The vegetation heterogeneity and consequent differences in rainfall interception and evaporation result in additional variability for simulated organic moss-humus layer moisture in dry conditions (see also Fig. 5). SAR and the SpaFHy-2D rootzone simulations agree on their main spatial patterns (i.e. drier forests and wetter peatlands). However, it is likely that SpaFHy-2D overestimates organic moss-humus layer moisture content variability, as there is a clear discrepancy between the SpaFHy-2D and the SAR-based estimates. The simulations provide too high moisture content in wet (Fig. 9A) and are biased low in drier (Fig. 9D) conditions. Compared to the simulations, SAR data shows significantly more cell-to-cell variability and the histogram appears nearly normally distributed, especially below 0.55 $m^3m^{-3}$ (mainly mineral soils).





Histograms of all daily soil moisture values in Fig. S3 confirm that the SAR data tends to be normally distributed between 0.1 and 0.5 $m^3 m^{-3}$. Considering the ability of SAR to well predict peatland soil moisture (Fig. 6), the agreement of SpaFHy-2D and SAR provide support for the earlier findings that soil moisture predictions improve when the lateral groundwater flow is included (SpaFHy-2D). A closer look at the rectangular box shown in Fig. 8 further confirms the good agreement of SpaFHy-2D -simulated and SAR-estimated rootzone moisture both at the dry and wet areas, but also highlights the high cell-to-cell variability in SAR-based soil moisture (Fig. 9).

### 3.5 Drivers of spatiotemporal soil moisture variability

To better separate the role of lateral groundwater flow and water table dynamics from that of vegetation heterogeneity under different temporal soil moisture regimes, Fig. 10 shows the grid-cell to grid-cell differences ($\Delta\theta$) between SpaFHy-2D and 1D simulations as well as between 1D and $1D_{homog.canopy}$ runs. The dry to wet conditions are represented by quantiles of 0.1, 0.5 and 0.9 of grid-cell rootzone soil moisture content. As expected, the difference is highest in wet conditions. In the wettest conditions (q = 0.9, Fig. 10C), the lateral groundwater flow has a large impact on soil moisture (mean $\Delta\theta$ between 2D and 1D ca. 0.1 $m^3 m^{-3}$) and in major part of the catchment, including also parts of the forested areas. The difference between the models is smallest at periods with intermediate soil moisture (q = 0.5, mean difference ca. 0.05 $m^3 m^{-3}$, Fig. 10B) when the differences emerge almost only in peatland grid-cells. Interestingly, the difference between the 2D and 1D predicted soil moisture is significant also in dry conditions (mean difference ca. 0.07 $m^3 m^{-3}$, Fig. 10A), indicating a long-lasting effect of lateral groundwater flow from the upland to the lowland grid-cells. The role of vegetation heterogeneity on soil moisture patterns is negligible at intermediate and wet conditions (Fig. 10E,F), and only minor differences are found in very dry conditions (Fig. 10D). The vegetation heterogeneity plays a larger role in the moisture content of the organic moss-humus layer, but the impact of lateral flow still remains stronger (Fig. S6).

## 4 Discussion

### 4.1 Insights on the role of lateral groundwater flow for top soil moisture

Our multi-scale data and high-resolution process-based simulations in the subarctic LJO catchment showed that regardless of the catchment hydrologic state (from very dry to very wet conditions), lateral groundwater flow plays a major role in shaping the spatial variability and dynamics of soil moisture (Fig. 10). The results indicate that spatially resolved models, which include groundwater flow are necessary to reliably predict the soil moisture variability at high-latitude catchments. Nevertheless, explicit description of lateral groundwater flow is commonly neglected in current hydrological and land surface models that operate at a coarse resolution (Best et al., 2011; Lawrence et al., 2012; Niu et al., 2011; Noilhan and Mahfouf, 1996). Increasing the spatial resolution of hydrological and biogeochemical land-surface models is the current trend; for instance Wood et al. (2011) set the ambition for future hyper-resolution LSMs to simulate at horizontal resolutions of 1 km for global-scale and 100 m for regional-scale. When the models are adapted to finer grids, it becomes increasingly important to implement

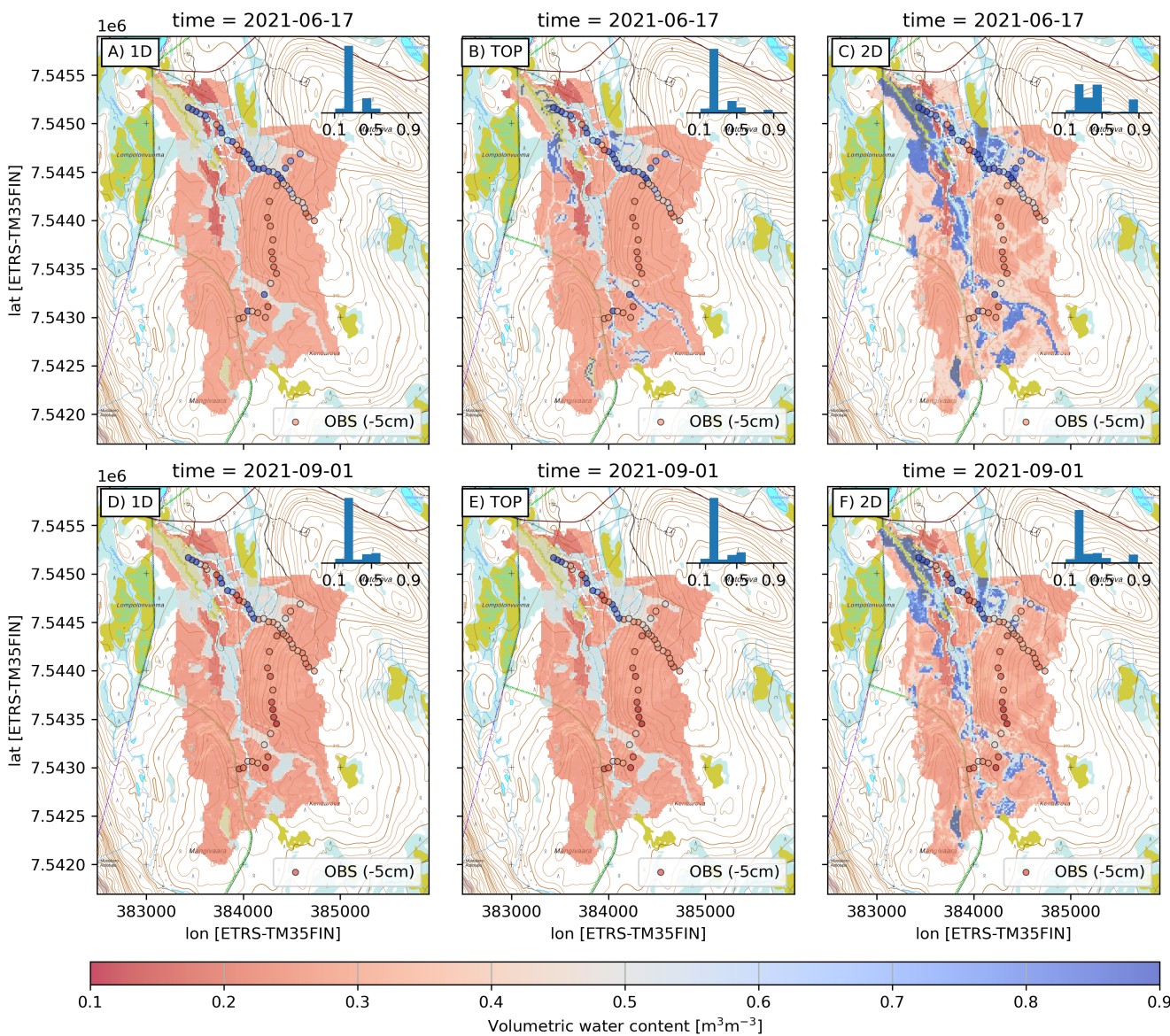

**Figure 7.** Spatial patters of modeled rootzone volumetric water content by the three model conceptualizations on 2021-06-17 (upper row, more moist) and 2021-08-01 (lower row, drier conditions). The bar plot shows binned distributions of simulated grid-cell soil moisture across the whole catchment, and in-situ measurements at 5 cm depth are shown as circles. The rasters overlay a topographic map (NLSF, 2020).



**Figure 8.** Spatial patters of SpaFHy-2D modeled rootzone and organic moss-humus moisture, and SAR-based estimates on wet (2019-06-26, upper row) and dry day (2019-08-01, lower row). In-situ measurements at 0 cm and 20 cm depths are shown as circles, and the bar plot shows binned distributions of simulated and SAR-estimated soil moisture across the whole catchment. The rectangular box shows an area that is presented in Fig. 9. The rasters overlay a topographic map (NLSF, 2020).





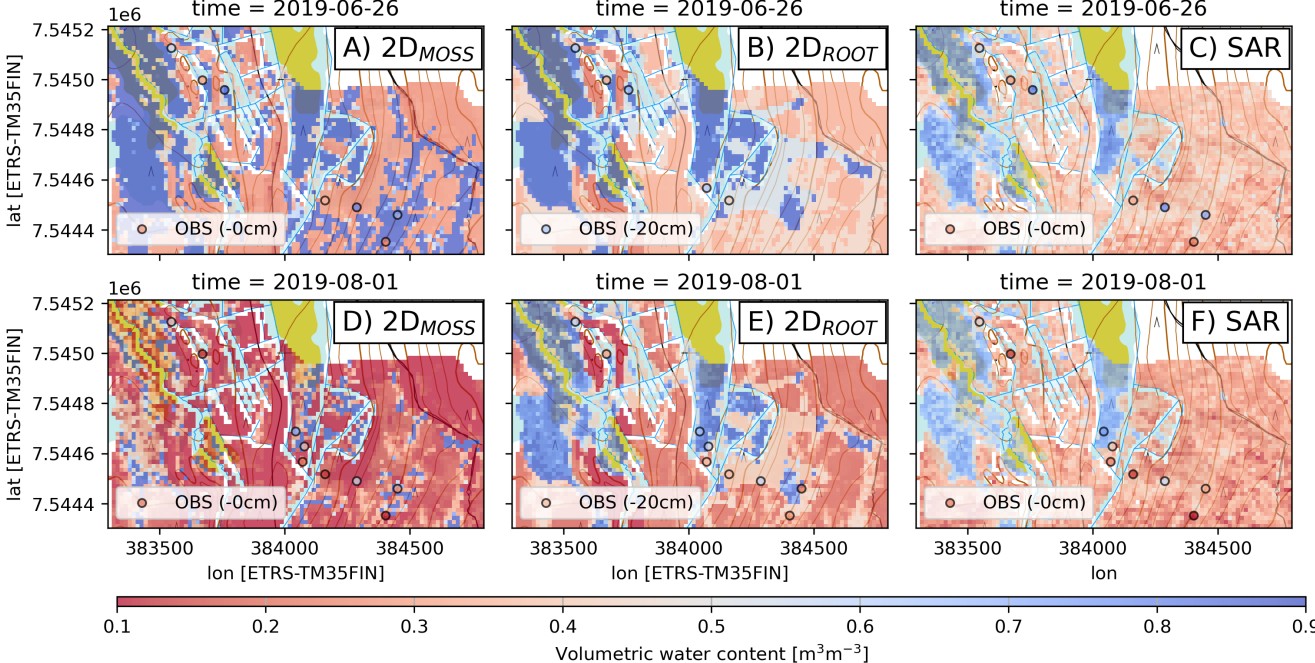

**Figure 9.** A zoomed in distribution of 2D modeled rootzone and organic moss-humus moisture and SAR-based estimates on wet (2019-06-26, upper row) and dry day (2019-08-01, lower row). In-situ measurements at 0 cm and 20 cm depths are shown as circles. The rasters overlay a topographic map (NLSF, 2020).

processes describing lateral groundwater flows (Ji et al., 2017; Kim and Mohanty, 2016; Decker et al., 2013). Our relatively simple 2D shallow groundwater flow model, incorporating only 7 additional parameters (water retention parameters, soil and

stream water level) which were determined using openly available digital elevation model, soil type and stream network rasters, performed comparably to the state-of-the-art integrated surface-groundwater model HydroGeoSphere (Brunner and Simmons, 2012) in predicting observed groundwater levels at the LJO catchment (see Fig. S7 and Fig. 6 in Autio et al. (2023)). Also, the simulation of groundwater influenced areas in the catchment is in broad agreement with the simulations by Autio et al. (2023).

Regardless of the known sensitivity of ecohydrological fluxes (i.e. interception, evaporation, transpiration) to changes in LAI and plant type (Launiainen et al., 2019; Kozii et al., 2020; Launiainen et al., 2016), the impact of lateral groundwater flow outweighed the impact of vegetation heterogeneity on soil moisture dynamics throughout the year. However, it is worth noting that the growing season in Pallas is rather short, and vegetation rather sparse and not very heterogeneous within the site classes (Fig. 2). In addition, the impact of vegetation heterogeneity on soil moisture is attenuated due to the compensating

processes; soil evaporation decreases while transpiration and interception evaporation increase with increasing LAI, resulting in less drastic changes in total ET (Leppä et al., 2020; Launiainen et al., 2019). Consistent with Kollet and Maxwell (2008), the impact of groundwater flow on top soil moisture persisted also in dry conditions, suggesting high resilience of low-lying cells

**Figure 10.** The impact of lateral groundwater flow (upper row) on rootzone soil moisture expressed as $\Delta\theta$ = 2D - 1D, and the impact of vegetation heterogeneity (bottom row) expressed as $\Delta$ = 1D - $1D_{homog.canopy}$ in different catchment soil moisture states. The panels correspond to 0.1, 0.5, and 0.9 quantiles of grid-cell soil moisture, and the bars show distribution of binned differences. Mean difference (MD) is shown in each panel.





to droughts due to long-lasting lateral flow from the upland cells. The simulations showed that at large parts of the catchment, rootzone moisture content was controlled by lateral groundwater flow, and the strength of the effect depends on the state of
the groundwater storage. Ji et al. (2017) showed that the role of lateral flux becomes crucial in high-resolution land surface simulations in a region dominated by a humid climate and coniferous forests in the western USA. At a resolution of 100 meters, they showed that subsurface lateral flow transports moisture from high elevation areas to valley bottoms, impacting local grid-cell and catchment average ET especially in dry conditions. Kollet and Maxwell (2008) coupled a groundwater and a land surface model, and demonstrated that when the water table depth was above 5 meters, there was strong coupling between
groundwater dynamics and land surface processes at the subhumid grassland-dominated watershed in the USA. Our results at LJO catchment are in line with these studies.

The impact of lateral flow was found especially important for peatlands, both due to the high porosity of peat (Menberu et al., 2021) and location in the valley bottom (Fig. 2,7,8). Mineral forest top soils can also be (temporarily) impacted by lateral flow, especially during heavy precipitation and snow melt, but the difference between 1D and 2D models remained smaller due
to the small difference in mineral soil field capacity and porosity (Fig. 10).

## 4.2 SAR-based soil moisture: Potential and limitations

The Sentinel-1 SAR-based soil moisture estimates were useful to supplement the point-scale in-situ measurements and confirm the plausibility of the spatial soil moisture predicted by the SpaFHy-2D. To this date, model developments and evaluations of soil moisture predictions in boreal and subarctic forests and peatlands have been typically limited to point-scale studies, which
fails to encompass the full spatiotemporal extent that distributed hydrological models are simulating (Launiainen et al., 2015; Ala-aho et al., 2017b; Tyystjärvi et al., 2022). We found SAR-estimates very useful for spatial model-data comparison, and envision SAR to have further potential, for instance, as a proxy for water table depth assimilation or improved estimates of topographic wetness indices (TWI, depth-to-water) in peatlands (Bechtold et al., 2020; Zhang et al., 2018).

The comparison between SAR-based estimates and modeled soil moisture was not straightforward and revealed limitations
in using the SAR-based data, for instance as ground-truth calibration data for hydrological models. A direct comparison is challenging due to the disparate penetration depth of SAR in soil (1–5 cm: Nolan and Fatland (2003)), contrasting with the model layering (rootzone layer of 0–30 cm). Indeed, the correspondence of SAR-based estimates against in-situ measurements in the rootzone (0–30 cm, Fig. 6) was poorer than the original validation of SAR-estimates against *in-situ* measurements at the surface soil (0 cm, Fig. 11 in Manninen et al. (2021)). Enhancing the comparability of hydrological models with SAR-estimates
would contribute to more effectively harnessing the SAR-based data. Another notable difference is that hydrological models, such as SpaFHy, treat individual grid-cells homogeneously, neglecting heterogeneity within the grid-cells, while SAR-estimates can integrate multiple backscattering signals for a given grid-cell (Manninen and Jaaskelainen, 2021). In turn, hydrological models can integrate temporal information, whereas SAR-based estimates are instantaneous.

We also emphasize the need for potential algorithm improvements in computing soil moisture from SAR signals. Given the
homogeneity of the vegetation and soil texture, some of the spatial variability in the SAR-based data appeared more as noise than realistic soil moisture patterns. SAR-estimated temporal dynamics were weak and partly hidden under the noise. The



different SAR incidence and view angles with respect to the topography, to a large extent, cause the systematic difference of the soil moisture estimates from the morning and evening flyover times (Fig. 4, Fig. 5). The topography-induced shading was markedly more challenging for the development of the evening soil moisture algorithm of SAR-based estimates (Manninen et al., 2021). The presence of different vegetation characteristics and soil textures further complicates the interpretation of the backscattering signals, leading to uncertainties and noise in soil moisture estimates. Overall, the capability of any remote sensing based soil moisture estimate to represent various meteorological and landscape conditions can only be as good as the training data. As acquiring high quality and representative in-situ soil moisture data is challenging and costly, we encourage deeper collaboration between hydrological measurements, modeling and remote sensing communities.

## 4.3 Model limitations and outlook

The modularity of SpaFHy (Launiainen et al., 2019) was ideal for comparing the impact of different conceptualizations of the lateral groundwater flow. Nevertheless, there are potentially relevant hydrological processes that are not yet represented. For instance, overland flow and soil freezing and thawing are currently omitted, and this may influence soil moisture dynamics, particularly during and after snowmelt and in the autumn (Ala-Aho et al., 2021). Lateral overland flow has been found to distribute water from saturated grid-cells to unsaturated areas (e.g. in subarctic tundra and boreal forests; Tang et al., 2014). We suspect that it may be especially important after snowmelt and heavy precipitation events on the low-lying flat peatlands of the catchment. The snowpack representation of SpaFHy successfully captured the snowmelt timing (Fig. S1, S4), but relies on a simple degree-day approach, potentially limiting its ability to fully capture snowmelt dynamics. Moreover, the radiation conditions on the forest floor within a specific grid-cell may be influenced by the surrounding forest canopy outside that grid-cell. Thus, employing 3D radiation transfer schemes that consider the shading from grid-cell neighbours (Webster et al., 2023), or the demography of individual trees within a grid-cell could be beneficial (Mazzotti et al., 2021). Although the 2D lateral groundwater flow module added process realism and significantly improved soil moisture simulations, it comes with a computational cost; in terms of running time the SpaFHy-2D is approximately sixty times slower than TOP and 1D versions. For instance, a one-year simulation with 1D and TOP was completed in 5.4 seconds, while 2D took 321.6 seconds. This can become a burden when applying the 2D model to large areas or when parameter calibration is needed.

Uncertainties in model simulations and model evaluation accumulate from multiple sources: input data, model parameters, model structure and errors in *in-situ* measurements (Moges et al., 2021). The meteorological forcing timeseries was constructed from observations at the upland forest site, and radiation data gaps were filled with ERA5 data (Hersbach et al., 2020). It is known that there are intrinsic uncertainties in meteorological observations (Stuefer et al., 2020). Although data gaps were limited, those filled by ERA5 data further add uncertainties in the model-data comparison (Raleigh et al., 2015). In addition, we used spatially uniform meteorological forcing (excluding radiation where topographic shading was accounted for) measured at the forest site that may have been slightly different to those experienced on the lowland peatlands (Aurela et al., 2015).

The model was initiated and parameterized based on the best available open geospatial data on the landscape characteristics. As Härkönen et al. (2015) found a good agreement between the mNFI-based and ground-based LAI estimates, and soil moisture patterns were not majorly altered by vegetation characteristics (Fig. 10), we assume that vegetation parameters did not create





marked biases in the soil moisture predictions. However, estimating soil hydraulic properties from available geospatial datasets is challenging (Launiainen et al., 2022) and can yield systematic uncertainties and biased local soil moisture. Modeling lateral groundwater flow by the proposed 2D Darcy scheme also requires distributed data on depth-to-bedrock. As such information was not readily available, these parameters were assigned as estimates.

## 510   5   Conclusions

We explored the controls of high-resolution soil moisture dynamics, particularly the role of lateral groundwater flow, in the sub-arctic Lompolojängänoja catchment. We combined soil moisture data from multiple sources, including in-situ measurements and Sentinel-1 SAR-based estimates, and interpreted soil moisture variability with high-resolution ($16 \times 16$ m$^2$) process-based hydrological modeling. To accomplish this, we extended the Spatial Forest Hydrology (SpaFHy) model with an explicit lateral
groundwater (2D Darcy flow) submodel, and compared it to existing approaches where lateral groundwater flow was either neglected (free drainage) or based on a simple TOPMODEL conceptualization. The results showed the major impact of lateral groundwater flow in shaping soil moisture dynamics, particularly post snowmelt and after heavy rainfall. The inclusion of the lateral groundwater flow model notably improved soil moisture simulations in forested peatlands and open peatlands. SAR-based soil moisture estimates were valuable in confirming modeled spatial patterns. Discrepancies in spatial resolutions, SAR
penetration depth, and model layering, however, hampered direct comparison. Moreover, the noise in SAR-based data, particularly under forested areas, complicates its use as ground-truth evaluation data for hydrological models. Our study provides novel insights and tools for predicting soil moisture dynamics at high-resolution, necessary for ecohydrological, biogeochemical, and climate change adaptation studies, as well as for land-use management and planning in high-latitude environments.

*Code and data availability.*   SpaFHy model version developed and used in this study is available at Nousu et al. (2024a). The code repository
also includes meteorological forcing files and geospatial input rasters.

In-situ hydrological measurement data, including soil moisture, evapotranspiration, groundwater levels, and specific discharge, are available at Nousu et al. (2024b).

*Author contributions.*   JPN, SL, PA and HM designed the research. JPN led the study and performed the model experiments and data analysis, with scientific contributions of KL, SL, GM, HM, TM and PA. In-situ manual measurements were conducted by JPN and MK. JPN was
responsible for writing the article, with contributions from all authors. MA and AL provided the energy flux data, and TM provided the SAR data. SL, HM, PA and AL were responsible for the funding acquisition.

*Competing interests.*   The contact author has declared that none of the authors has any competing interests.





*Acknowledgements.* This work was funded by the Research Council of Finland (RCF) (ArcI Profi 4). Giulia Mazzotti was funded by the Swiss National Science Foundation (grant no. P500PN_202741). Samuli Launiainen and Jari-Pekka Nousu acknowledge the GreenFeed-Back project from the EU Horizon Europe Framework Programme for Research and Innovation (grant no. 101056921). Samuli Launiainen acknowledges the support of RCF (no. 356138 & 348102). Pertti Ala-aho was funded by the RCF Research Fellow grant (no. 347348). We acknowledge the Ministry of Transport and Communications through the Integrated Carbon Observing System (ICOS), ICOS Finland and RCF (grant no. 308511). The authors would like to thank Emmihenna Jääskeläinen and Anna Autio for valuable discussions during this work. We also acknowledge the use of ChatGPT 3.5 (Open AI, https://chat.openai.com) to proofread parts of the paper.





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
