# Peer review of "Multi-scale soil moisture data and process-based modeling reveal the importance of lateral groundwater flow in a subarctic catchment"

_Hydrology and Earth System Sciences, 2024_

## Referee Comment (RC3)

[referee-annotated manuscript omitted]

---

## Author Response (AR1)

**Author's response**

Black: reviewer comments
Blue: Author replies during interactive discussion
Red: New author replies during revision.

Thank you again to all the referees and the editor for your reviews, comments, and suggestions concerning our manuscript. We have revised the manuscript based on all your feedback and our earlier comments. Note that all planned revisions have been conducted in text/figures/tables as indicated (or slightly reworded) in our replies during the interactive discussion (blue text), unless otherwise clarified here (in red).

Additionally, we have gone through the entire manuscript to improve the quality of English language and re-organized parts of results and discussion to reduce overlap and improve the clarity of the manuscript. We have moved part of the SAR related text from Sects. 2.6 and 3.3 to Sects. 3.4 and 4.2. We believe this clarifies the analysis and discussion concerning SAR data. We have also made Sect. 3.4 more quantitative and added supportive tables and figures to the supplement. The results and conclusions of the study remain the same despite these changes. The constructive reviewer comments and our changes have substantially improved the manuscript. Please find our additional clarifications below in red.

**Author's reply to Referee #1**

General comments:

The paper addresses research questions which are both important and relevant to the scope of HESS. Through comparing several modelling approaches with different types of data, this study presents novel ideas and data to demonstrate the importance of including lateral groundwater flow in models representing soil moisture, particularly when simulating saturated peatland conditions and lowland forest. These results are significant given many models do not currently include lateral groundwater flow, and the importance of accurate simulation of soil moisture. This is of particular importance for peatlands, given their role in mitigation of climate change impacts.

The paper is well-structured and written clearly and concisely. The abstract summarises the research well and the introduction reviews the current gaps in the literature, explaining the need for this research. Methods and assumptions are clearly defined, and the results support the conclusions with well-presented figures and analysis. Limitations to the study are well-described and supplementary materials clear.

I have a few specific comments, but otherwise an excellent paper.

We would like to thank the reviewer for the positive feedback on our work. Please find our replies to each specific comment in blue.

Specific comments:

Abstract - Could highlight why this research is needed within the first couple of sentences.

Good idea, the beginning of the abstract will be revised as: "Soil moisture plays a key role in soil nutrient and carbon cycling, plant productivity and in energy, water, and greenhouse gas exchanges between the land and the atmosphere. However, the knowledge on drivers of spatiotemporal soil moisture dynamics in subarctic landscape is limited. In this study, we used the Spatial Forest Hydrology (SpaFHy) model, in-situ soil moisture data and Sentinel-1 SAR-based soil moisture estimates to explore spatiotemporal controls of soil moisture in a subarctic headwater catchment in northwestern Finland."

Please find revised abstract in the revised manuscript.

Line 104: "at range" –> across/over a range

Thanks, this will be corrected.

Line 116: plant Latin names in italics normally?

Indeed, these will be changed to italic.

Figure 1. It's not very clear how a) fits into b) and the scale of b) compared to c). It would also be nice in figure 1a) to see which is the forest site and which is the mire site (perhaps clarify in the figure legend in brackets). Perhaps also add the river network over the top so this is clear.

Note that the catchment is very small (see the scale in Fig. 1a), and therefore it would not be at all visible if plotted in Fig. 1b. We will add scale or coordinates to Fig. 1b so that it can be better compared with Fig. 1c.

We are going to add (forest) and (open mire) either in the figure legend or in the caption.

Note that the stream network is shown in Fig. 2: streams. We are also going to add the stream network in Fig. 1.

Please find revised Fig. 1 and caption in the revised manuscript.

Line 144: I assume the above-ground fluxes and state variables computed in the canopy submodel are the same for all 3 versions of SpaFHy? I would just add a sentence to make sure this is clear.

Yes, and this is already mentioned in the model parameterization section at L218 We will also add this earlier in the model description at L152: "The canopy and bucket submodels are common to all three SpaFHy versions.".

The new sentence can be found at L157 in the revised manuscript.

Line 227: Did you do any sensitivity tests to confirm that depth-to-bedrock was not important? Or was 5m decided based on any other studies/references?

Reliable and high-resolution information on depth-to-bedrock or the quality of the bedrock (e.g. porosity) is very difficult to obtain. At Pallas, there are some, yet very limited measurements available. Hence, we opted for a pragmatic approach, to assign a general value of depth-to-bedrock that approximately corresponds to both the thickest peat layers of the catchment, and the shallowest mineral soil depths found in the uplands of the catchment. No sensitivity tests were conducted. We will add following texts and reference to a published groundwater model study by Autio et al. 2023, providing context for the selected depth-to-bedrock values.

L228: "Due to a lack of reliable data on the depth-to-bedrock, a uniform thickness of 5 m was assigned for the deep soil layer of the 2D groundwater module throughout the model domain. This estimate corresponds approximately to the thickest peat layers and the shallowest mineral soil depths at the catchment (Autio et al., 2023)."

Please find the revised description starting at L236 in the revised manuscript.

L228: "No further calibration or sensitivity tests of any model parameters were conducted in this study."

See L240 in the revised manuscript.

Line 242/Figure 1: Not very clear which soil moisture sampling locations are those bi-weekly vs the 56 additional locations in the figure. I can also see there are locations labelled with m and l, but you don't explain what these two stand for.

The bi-weekly measurements are denoted with "i". The "l" and "m" correspond to those measurements that were only conducted twice (2021-06-17 and 2021-09-01). "l" measurements are conducted on a snow survey transect and "m" is an additional transect in different direction. These will be mentioned in the revised manuscript:

L242: "In particular, during snow-free seasons, biweekly manual measurements at 15 different points (denoted as "i" in Fig. 1) were conducted"

See L252 in the revised manuscript.

L244: "Additionally, we conducted two extended soil moisture measurement campaigns, including 56 additional locations (denoted as "l" and "m" in Fig. 1)"

See L254 in the revised manuscript.

Line 269: "of midday" -> at or for midday

Thanks, will be corrected.

Table 2: Figure legend above table rather than below

Legends of Table 1 and 2 will be shifted above the tables.

All table captions have been shifted.

Figure 3: Please add in the figure legend for A) as observed (obs) and simulated (mod). In addition, there is no SWE in the figure. For B) and C) figure legend I think as you have specified what each parameter is in A, it makes sense to do so here too i.e. Air temperature (T)

These will be added in the figure caption.

Line 313: "groundwater rechange" -> groundwater recharge

This typo will be corrected.

Figure 5. figure legend – add "(obs)" for "in-situ measured"

This will be added.

Figure 6: as previous plots have low=dark blue, high = yellow, this should be consistent for canopy fraction too

Good point, the colormap will be reversed here.

Please find the new Fig. 6 in the revised manuscript.

Line 384: "follow mostly" -> mostly follow

This will be revised.

Line 508: I assume there was no groundwater level data to compare with model simulations.

We have compared the simulations to the groundwater level data in the Supplement (Fig. S7). This was mentioned at lines 344-346 in the manuscript. We will reiterate this at L509: "Even with these limitations, the modeled groundwater level dynamics were relatively close to the observed levels (Fig. S7)"

See L535 in the revised manuscript.

**Author's reply to Referee #2**

I read this manuscript describing a modelling investigation of lateral groundwater flow in a subarctic catchment with interest. The authors use multiple model parameterizations to quantify hydrologic fluxes and states in a sub-Arctic catchment, with a focus on soil moisture. The results point to the importance of model inclusion of lateral groundwater to improve simulation of soil moisture, with the SpaFHy-2D set-up generating a greater range in soil moisture conditions than the other two parameterizations, most notably increasing moisture levels for peatland soils, which are expected to have higher saturation. Overall I think the work has merit, is suited to HESS, and can be of interest to the readership of the journal with revisions. I found the paper to generally well written, especially the introduction, which provides the reader with a generally clear picture of the research needs and direction. I do however highlight below a number of areas concern, including a caution in how the model results have been evaluated.

The model, despite the potential afforded by the new parameterization approach described here, is nonetheless not demonstrating particularly strong ability to simulate soil moisture, and a more rounded presentation of model performance in analyzing the results is merited. The authors point to the potential for enhanced process representation as being potentially needed, which I would agree with as cold-regions processes, including snowmelt dynamics, soil freezing and infiltration inhibition, and more are not yet considered in the model. Here I summarize some of the overarching areas for improvement that I suggest for this work.

We would like to thank the reviewer for the careful review and constructive feedback. Please find our replies to each comment below in blue.

We recognize the need for better implementation of these cold regions processes especially for studies focusing on wintertime and are planning to implement them in the future version of SpaFHy. However, we believe their impact for (mostly) summertime soil moisture is not major, and implementing these processes were out of scope for this particular study. These model limitations are discussed in Sect. 4.3.

1. In the methods, additional detail on what processes are captured and how they are represented in the model are warranted. How are precipitation and snowmelt represented to characterize hydrological dynamics of upper layer? Does water percolating through the soil bucket immediately reach the catchment outlet once it drains below this layer, or does the model account for transit time to the outlet, and if so how is flow routing handled? How are snowmelt dynamics modelled? Do frozen soils prevent infiltration?

Thank you for pointing these out. We are going to revise the text by citing to Launiainen et al. 2019 model description section, but also by explaining the approaches that solve relevant processes for this study at L144: "The above-ground fluxes and state variables are computed in the canopy submodel, including rainfall and snowfall interception and evaporation, throughfall, transpiration, and snow accumulation and snowmelt (see Sect. 2.2. in Launiainen et al. 2019). Snowmelt is computed with a degree-day approach while ET components are solved by the Penman–Monteith equation. For transpiration, the canopy conductance is derived from accounting for the stomatal optimality principle and exponential attenuation of light in the canopy (Launiainen et al. 2019)."

Indeed, information about the discharge was vague. The discharge is assumed to immediately reach catchment outlet (no transit time). We will clarify this better in the revised manuscript:
L149: "The lateral water flow between the grid-cells is omitted, and drainage from the bucket submodel is removed from the model domain as stream discharge at the catchment outlet without delay."
L163: "Catchment discharge at the catchment outlet is the sum of the catchment average baseflow (predicted by TOPMODEL) and surface runoff without delay."
L183: "We do not consider temporal changes in stream water level and omit channel flow in the stream network; thus the sum of the outflow into the stream cells and surface runoff form the runoff at the catchment outlet without delay."

See the new Sect. 2.2.1 – 2.2.3 in the revised manuscript.

Soil freezing is not simulated and hence does not affect simulated infiltration. This is already discussed in the model limitation at L483. According to preliminary and unprocessed site data from the past few years, the soil freezing is rather minimal at the site, presumably due to the early and deep snow cover. Hence, we believe that assuming soil freezing does not significantly impact catchment soil moisture is justified, particularly during the growing season.

Section 2.5 While there are a wealth of soil moisture measurements observations described here, it is not clear how these correspond to the modelling framework. Which depths represent the organic layer, and which depths represent the rooting zone? Section 2.4 makes effort to descript the horizontal resolution of the

modelling, but the vertical resolution has not been described. This detail is important in linking observational data to the modelling approach, and a fuller explanation here would be most helpful.

Thank you, that is a good point. The soil moisture measurements were conducted from the soil surface (0 cm) to 30 cm depth, so practically they all represent the rootzone layer. While organic layer moisture was not directly measured, the measurement at 0 cm depth is closest to the modelled organic moss-humus layer (impacted by soil evaporation). We will clarify this in the revised manuscript at L257: "All soil moisture measurements from 0 to 30 cm depth correspond to the rootzone layer of SpaFHy. Although moisture of the organic moss-humus layer was not directly measured, the measurements at the soil surface (0 cm depth) can, to some extent, represent the dynamics in this layer (e.g. impacted by ground evaporation)."

We will also further stress in the discussion that comparison of modelled and in-situ measured, and SAR-based soil moisture is complicated as they represent different depths or be integrated measures of water content. This vertical mismatch is common hindrance in comparing model results with data (e.g. Tyystjärvi et al., 2021; Shellito et al., 2020). This will be revised at L464 as: "This vertical mismatch is a common challenge (Shellito et al., 2020), and hence, enhancing the comparability of in-situ measurement as well as hydrological models with SAR-estimates would contribute to more effectively harnessing the SAR-based data."

Please find the revised description at L269 and L478 in the revised manuscript.

2. In the results, there is a pattern of the results paragraphs starting with pseudo figure captions rather than topic sentences. As a result, it is not easy for the reader to easily parse out key themes from the analysis conducted. These sometimes appear later in the paragraphs as well. Removing these would improve conciseness and provide a more direct overview of the findings emerging from this work.

Thank you for suggesting these improvements. We are going to revise the manuscript accordingly.

Please see the revised results section in the revised manuscript.

3. In the results, there is need for a more systematic approach to presenting evaluation metrics. While overall metrics are presented for some of the model fits, the results describe permutations of these relationships that are not quantified in the text. Importantly, some of the description of the model performance is not well supported by the analyses shown (see detailed examples below). That the model, even with 2D representation, does not perform particularly well should be emphasized, as there is lots of direction provided in the discussion about how this work could be improved in the

future. It is important that the model performance, and its limited ability to capture soil moisture dynamics be described with appropriate supporting quantitative metrics.

Thank you for these comments. You're right that even the improved soil moisture simulations by the 2D approach are far from perfect. Nevertheless, the aim of this paper was not to calibrate the model to obtain optimized soil moisture simulations at this particular catchment, or at those point-observation locations. Instead, our goal was to identify potential areas that are impacted by lateral groundwater flow, and to develop a simple enough modelling approach that can capture the impact of groundwater dynamics, and which could be applied to other Finnish or boreal catchments with less site information.

The main limitation in the model performance is likely due to the uncertainty in soil hydraulic parameters, as they cannot be directly measured spatially or be well predicted from available spatial datasets (see e.g. Launiainen et al. 2022). As shown by the study, the simulated volumetric soil moisture is strongly affected by soil hydraulic properties that have (in reality) high spatial variability, and hence, high uncertainties when estimated from geospatial data. For instance, Launiainen et al. 2022 have shown that it is difficult to predict soil hydraulic properties from available geospatial data, such as soil texture maps.

We do recognize that the 2D modelling approach could be further improved as it tends to simulate more saturated areas compared to the data (as discussed in the manuscript). However, more developments are needed to produce accurate soil maps that spatially distributed modelling approaches can utilize.

We are going to revise the manuscript to elaborate on the limitations in the soil parameterization, model performance compared to observations - and to further stress that the goal was not to obtain 1:1 match with the data.

We have revised the results and discussion sections to clarify that the improved soil moisture simulations by SpaFHy-2D still suffer from uncertainties. Additionally, we have further emphasized the aims of this study in the revised manuscript.

L319: "The model overestimates the rootzone soil moisture content and its temporal change compared to the mean of point-scale observations in the rootzone at Kenttärova (MBE: 0.05 $m^3m^{-3}$). This mismatch is likely due to the uncertainties in soil hydraulic parameters that could potentially be corrected by calibrating soil field capacity and wilting point. However, as the simulations mostly fall within the observed range (MBE: -0.01 $m^3m^{-3}$ when compared to observed maximum) and this comparison only represents one grid-cell, such calibration was not considered meaningful for the aims of this study."

Please find a revised text starting at L323 in the revised manuscript.

L336: "Minor discrepancies between SpaFHy-2D predicted rootzone and in-situ measured soil moisture content are likely due to uncertainties in soil hydraulic parameters (e.g. an overestimation of field capacity in Fig. 5 i14)."

Please find the revised text starting at L329 in the revised manuscript.

L355: "All evaluation metrics are considerably better for the 2D model, but it tends to overestimate soil moisture on the peatland grid-cells, consistent with the overestimation in Fig 5F,G, hence the performance is still limited. Nevertheless, the aim of this study is to assess the influence of groundwater and lateral flow on shallow soil moisture dynamics, rather than producing calibrated soil moisture simulations, so the performance of the 2D model is considered satisfactory."

See L357 and L338 in the revised manuscript.

L491: "Although the 2D lateral groundwater flow module added process realism, the soil moisture simulations were still only satisfactory due to uncertainties in classifying soil types and because soil moisture data was not used to calibrate the model's soil hydraulic parameters. However, as the aim of the study was not to produce calibrated soil moisture simulations, these uncertainties and deviations from observations are considered acceptable. Additionally, the 2D model came with a computational cost; in terms of running time the SpaFHy-2D is approximately sixty times slower than TOP and 1D versions."

See L516 in the revised manuscript.

In addition, we are going to add performance metrics wherever suitable (see replies to specific comments below concerning e.g. groundwater levels, Fig. 6 and Sect. 3.4.: SAR and SpaFHy comparison)

4. While I understand the intent behind including the SAR data, my assessment is that they are being relied on too heavily in this work. Given that the SAR data do not capture very well other observations of soil moisture, e.g. due to spatial differences in representivity, there is limited potential in using them to assess model performance. This is an area where the paper can be streamlined, perhaps by earmarking some of this for the SI rather than the main text.

We do not mean to rely on the SAR data too heavily and use it more qualitatively than quantitatively. The SAR-based soil moisture maps serve as an independent

estimate of soil moisture covering a larger area than any of the point measurements. Comparison of the different model versions and SAR based soil moisture maps helps in locating areas of agreement/disagreement for the different approaches (Figs. 7 and 8). As demonstrated by Fig. 6D, the skill of SAR data to estimate points that have groundwater influence is sufficient for the aims of this study (e.g. peat soils with soil moisture > 0.55 $m^3m^{-3}$). Our comparison show that the methods have areas of agreement but also where they disagree, and these results can further aid in development of both soil moisture estimation methods.

It is also worth noting that Referee #1 and #3 were satisfied using the SAR data in the manuscript, and even so that Referee #3 asked for extending the analysis by providing quantitative evaluation between the model and SAR. Hence, we prefer to keep the analysis with SAR data in the main manuscript as we believe it brings value for the study.

In the revised manuscript, we moved part of the SAR related analysis from Sects. 2.6 and 3.3 to Sects. 3.4 and 4.2. We believe this clarifies the manuscript, intended use of SAR and improves the discussion concerning SAR data.

Line comments:

Introduction

L23: "region, climate change"

Thank you, will be corrected.

L27: "affecting tree health, mortality"

Thanks, will be corrected.

L33: Here it might be helpful to expand on the C aspects a bit. This are of research is important for understanding GHG exchange from terrestrial environments, but also lateral export of DOC (and nutrients, as noted) to waterbodies.

This will be revised as:
"Hence, accurate information on spatiotemporal soil moisture conditions has the potential to improve estimates of tree health, terrestial carbon stocks and greenhouse gas sinks and sources, and lateral export and leaching of carbon and nutrients (Bond-Lamberty et al., 2016; Nakhavali et al., 2021)."

Please see the revised introduction.

L41: I wonder if undulation is the best term to use here. I tend to think about undulating surfaces as occurring over space, but for groundwater it seems implied here that this is temporal variability in groundwater height at a given location, rather than an undulating groundwater surface that rises up and down repeatedly along a linear plane.

Good point, we will change this to "variation in the water table depth"

L58: ", and extend point-scale…"

Thanks, this will be corrected.

L93: perhaps use "shallow soil moisture" here. Also, I think you can simplify this question to one of 'where', since, as worded, it seeks to look at temporal variability. In my mind, this makes the 'when' part of this question redundant.

You're right, shallow soil moisture is better, and will be corrected. We agree that the 'when' and 'temporal' are redundant, it will be revised as: "Where does lateral groundwater flow affect the temporal variability of shallow soil moisture?"

L96: perhaps it is worth adding here an investigation of the accuracy of the SAR measurements using point scale observations. This seems a prerequisite to using soil moisture estimates to evaluate model predictions.

Although we do some investigations of the accuracy of the SAR data, we do not believe that it should be stressed among the scientific questions of our study. The SAR soil moisture data has been already evaluated in the original publication by Manninen et al., 2021.

Methods

L107: can you include the proportion of precipitation falling as snow? This would be helpful.

We are going to add the proportion of snow as: "The proportion of precipitation falling as snow is approximately 42 % (Marttila et al., 2021), and the seasonal snow cover persists from about October until May (Aurela et al., 2015)."

L109: use elevation instead of altitude (which generally refers to height above the ground surface).

Elevation will be used, thanks.

L111: the image and text below suggest roads and ditches as potential human disturbances. Perhaps the extent of human disturbance (while still small) could be described in more detail here.

You're right. We will revise this as: "Except for a few small roads and ditches, the area has had little human influence and can be considered mostly a pristine subarctic headwater catchment."

L118: Above (L116) the scientific name is used, but here the common name appears. Suggest defining first and using standardized naming convention for all plant species. Perhaps the journal has a convention for this.

Thank you, this will be corrected.

L129: What soil depth is used/modelled here to represent the rooting zone?

Rootzone depth of 0.3 m was used. This can be found in Table 2. We are also going to add sentences to mention both the organic moss-humus layer and rootzone layer depths in Sect. 2.4. as follows,
L221: "A depth of 0.05 m was assigned to the organic moss-humus layer."
L227: "The rootzone was assigned a depth of 0.30 m."

Please find the above-mentioned changes at L228 and L235 in the revised manuscript.

L147: It seems strange that precipitation and snowmelt are not also represented here to characterize hydrological dynamics of this upper layer? Does water percolating through the soil bucket immediately reach the catchment outlet once it drains below this layer, or does the model account for transit time to the outlet? How are snowmelt dynamics modelled?

Precipitation and snowmelt are represented in the canopy submodel. The upper organic moss-humus layer receives throughfall (depends on canopy submodel interception, evaporation and throughfall). This will be revised at L146: "The upper layer is the organic moss-humus layer, whose water budget is affected by throughfall interception and evaporation, as well as infiltration to the lower rootzone layer, where drainage and transpiration take place."

See L150 in the revised manuscript.

Regarding water percolation and snowmelt, see our answer for one of the first comments. In the 1D version, all water draining from the rootzone soil bucket becomes immediately runoff at the catchment outlet. In the TOP-version (TOPMODEL), the water is routed through conceptual groundwater storage from which it becomes either baseflow to catchment outlet, or saturated overland flow at

specific grid-cells (without any delay). In the 2D version, the shallow groundwater flow is explicitly simulated from grid-cell to grid-cell using the Darcy law and becomes flow to ditch at the ditch nodes. No routing of overland flow or channel flow was considered. Overland flow and flow to ditch becomes runoff at the catchment outlet with no delay.

L181: Can you provide more detail on constant h in streams and ditches, as this seems to be a strange assumption to make as these would be dynamic in time and space. Perhaps these are not spatially explicit in the model, but the reader would benefit from a fuller description here, as it is not clear how catchment discharge is captured if streams have constant h.

The assumption to keep constant stream water level (h) simplifies the modelling framework as we do not need to explicitly represent surface water dynamics (channel flow) nor to couple them to groundwater dynamics.

We assume that stream water levels do not have significant impact on the catchment soil moisture dynamics. The stream water level dynamics have likely impact on the soil moisture adjacent to the streams and discharge dynamics, but neither was the focus of this study. Yet, assuming constant stream water level at -0.2 m below the surrounding environment resulted in satisfactory discharge simulations in this study. There are other models that have integrated surface-groundwater dynamics (e.g. HydroGeoSphere, Autio et al., 2023), explicitly simulating the difference in hydraulic head between the groundwater and stream, and its small-scale influence on soil moisture. For our study, adding channel flow was considered out of scope.

We are going add at L184: "The assumption of a constant stream water level simplifies the modeling framework and should not significantly impact catchment soil moisture dynamics."

See L189 in the revised manuscript.

L242: During what time period were bi-weekly observations made, did this span the full calendar year?

Thanks, this needs clarification. Only snow-free season was measured:
Will be revised as: "In particular, during snow-free seasons, biweekly manual measurements at 15 different points were conducted using WET-2 and PR2 Profile Probe sensors with an HH2 readout unit (Delta-T Devices Ltd., Cambridge, U.K.) sampling soil moisture profile at depths 0 cm, 10 cm, 20 cm and 30 cm (from the soil surface)."

L249: Porosity provided here (0.88) does not match that in table S2 (0.89)?

Thank you, this typo will be corrected.

L278: Is 'matching' meant here instead of 'non matching', the context of this description seems to be characterizing the data available to use, rather than the inverse(?).

This was not clear. It was supposed to mean that there were measurements (in Manninen et al. 2021) that did not match the SAR overpass time. Will be revised as: "Yet, the validity of the method could be checked only at 92 different locations, for which soil moisture data was available: altogether there were 678 discrete values and eight points of continuous data. The discrete soil moisture measurements did not match the overpass times of SAR, which complicated the development of the soil moisture estimation method. Naturally, for validation the soil moisture estimates were calculated for the times matching the individual measurements. Hence, the time difference complicated the method retrieval, but not the validation."

After revisiting this section, we decided to remove altogether this text from the manuscript as this was already addressed and discussed by Manninen et al. 2021 and is not essential for this particular study. We have also moved parts of the discussive SAR description to Sect. 4.2. for clarity.

Results

L307: Why are different evaluation metrics being used for Q and ET?

Selection of evaluation metrics is a subjective choice, and using different metric for different variables is not unusual (Ala-aho et al. 2017, Duoinot et al 2019, Launiainen et al 2019). In our case, the validation data are considerably different; for instance, ET data have missing values and manual soil moisture measurements are sporadic, while discharge data are continuous. Thus, using the same metric could misleadingly suggest that it is reasonable to compare model performance across these different variables, which is not justified. Moreover, metrics such as the Kling-Gupta efficiency (KGE) are commonly used to evaluate discharge simulations against observations (see e.g. Knoben et al., 2019)

L325: It would be helpful to explain here why the morning flyover predicts drier soil conditions. It seems that this pattern could depend on the time of year, and whether snowmelt is occurring during the day.

We agree that such insights are interesting, but we consider that additional analysis on SAR-data are out of scope of this study. The morning-evening flyover difference was considered by Manninen et al. 2021 in their SAR soil moisture paper. The difference between morning and evening algorithm quality is very much related to the terrain. In Finland, due to the ice era, the hilltop line of the terrain is typically in northwest-southeast direction very much in line with the evening pass orbit of

Sentinel-1. For this reason, the evening images tend to have more serious shadowing (in the distal slopes) than the morning images. Inevitably this leads to poorer quality of the soil moisture estimates. We will guide the reader to Manninen et al., 2021 at L326: "Further discussion on the differences between these SAR flyovers can be found in Manninen et al. 2021."

We have added this in discussion at L491.

L335: Please explain what is meant by "especially in terms of ranking the positions". It seems clear from the results that some locations are captured well, and others poorly.

We mean that the simulations capture the wet peatland locations at or near saturation (theta > 0.7 $m^3m^{-3}$: i7, i8), the intermediate mixed locations (0.5 < theta < 0.7 $m^3m^{-3}$: i16, i20), and the dry forest locations (< 0.5 $m^3m^{-3}$: i14, i18) reasonably well. Certainly, there are also locations where the simulations do not match the observed range for most of the season (e.g. i21, i17).

We will clarify this in the text as: "The SpaFHy-2D predicts the rootzone soil moisture differences between the locations reasonably well, especially in terms of ranking the locations between wet, intermediate and dry moisture conditions (Fig. 5)"

L337: Yes, but there are also extended periods of strong underestimation that should not be overlooked.

Yes. Will be revised as: "The SAR-based soil moisture lacks the observed temporal variation whereas SpaFHy-2D simulations tend to overestimate temporal dynamics compared to the in-situ observations. The SAR-based estimates also noticeably underestimate mean soil moisture content."

See the revised text starting at L342.

L345: Some metrics should be provided on all of these evaluations.

Note that we do not consider groundwater levels in and above the rootzone (i.e. highest groundwater level is -0.3 m). We are going to add the following groundwater level performance metrics table in the Supplement:

**Table 4.** Performance metrics of 2D simulated and in-situ measured groundwater levels at 10 different locations around the catchment.

| Location | Mean biased error (m) | Mean absolute error (m) |
|----------|----------------------|-------------------------|
| PVP1 | 1.04 | 1.07 |
| PVP2 | -0.08 | 0.15 |
| PVP4 | 0.62 | 0.67 |
| PVP5 | 0.61 | 0.63 |
| PVP7 | 1.27 | 1.29 |
| PVP8 | 2.34 | 2.34 |
| PVP9 | -0.70 | 0.79 |
| PVP11 | -0.81 | 0.86 |
| PZ1 | -0.51 | 0.51 |
| PZ2 | -0.37 | 0.37 |
| MEAN | 0.34 | 0.87 |

Please see Table S4 in supplement of the revised manuscript.

L350: I don't believe that this statement is supported by the data. R2 values are low. There are commonly large absolute errors approaching 0.5 m³ m⁻³ at the upper end of this range. Large relative errors at lower observed soil moisture levels and a tendency to overpredict is clear. Perhaps use of NMAE would offer a better assessment here, but importantly, the model abilities should be described with greater rigour.

See our earlier replies regarding the high uncertainties in soil hydraulic parameters. In fact, this is a good example, as most of these large absolute errors (approaching 0.5 $m^3$ $m^{-3}$) are due to wrong soil parameterization: observed soil moisture content suggest (apparently very high porosity) that the soil is likely organic peat while in the model (based on geospatial soil data) the grid-cell is parameterized as mineral soil.

We will revise the manuscript by discussing more about the model limitations, and revise this particular statement as: "The observed soil moisture below ca. 0.55 $m^3$ $m^-$$^3$ are rather well captured by all model conceptualizations, especially considering the uncertainties in soil hydraulic parameters based on geospatial data (Fig. 2: soil type)"

L353: Again here, a more objective description of the model performance is required. There are few predictions at higher moisture with the 2D model that lie close to the 1 to 1 line. If 0.55 is used as a threshold for evaluating performance, it is recommended that metrics for observed moisture levels above and below this level, as well as overall be presented. Likewise, it would be helpful to provide metrics for the different landcover or canopy closures discussed in the text. Many of the examples in the following paragraph relate to landcover, so this seems a better factor to use in Figure 6 than is canopy closure.

Thank you for this comment. We agree that considering site or soil types in addition to canopy closure is interesting. They do however contain overlapping information (low canopy fraction = peatland, high canopy fraction = mineral forest soil). We have made an alternative version of Fig. 6 with the dominating soil types (peat and

mineral soil) and calculated separate metrics. This new figure will be added in the supplement of the revised manuscript (see below). It confirms the findings from the original Fig. 6, but also highlights the uncertainties in soil parameterization based on the geospatial data (soiltype): many of the peat soil points of the 2D approach are overestimated (some also underestimated). As noted before, we will further highlight the uncertainties in soil parameters in the revised manuscript.

[Figure]

*Figure 1. Comparison of simulated rootzone soil moisture content and SAR-based surface soil moisture estimates against spatiotemporal manual in-situ soil moisture observations. The blue color of the points correspond to peat soil and the orange color to mineral soil.*

See Fig. S6 in supplement of the revised manuscript and text starting at L360.

L370: It would be helpful to cite Figure 6 here in addition to mentioning this occurs in peatlands.

Good point, will be added.

L374: Again here, there remain large deviations with this model, and care should be taken to not oversell what the model is capable of.

Indeed, we will specify that the model matches well the point observations shown in Fig. 7.

L380: Perhaps I have misunderstood something, but I am having trouble understanding the value in comparing the model to SAR measurements for a 5 cm depth with modelled data, given that those SAR measurements haven't been validated as being in strong agreement with observed data (Figure 6D). I appreciate that the SAR data provide an opportunity to compare model results between observations, but this would only seem useful if those SAR data are effectively capturing the hydrological state, and this has not been shown clearly. For this reason, I am unconvinced that section 3.4 belongs in the manuscript. The discussion beginning at lines 469 seems to support this notion.

We do recognize the limitations in using the SAR-based data, as discussed in Sect. 4.2., but we argue that the added value of including them is greater.

First of all, Manninen et al. 2021 have shown that the SAR based soil moisture estimates are relatively accurate when evaluated against measurements of matching times and soil depths. Of course, in our case, the absolute skill of SAR is poorer as we are comparing measurements that are also beyond the penetration depth (rootzone) or exact overpass times. In addition, we have resampled the SAR data to match the simulated grid. In addition, the SAR based soil moisture estimates are instantaneous midday values, but the model-based estimates are diurnal averages. For that reason, exact fit between these two soil moisture estimates can't be expected, even if both estimation methods were perfect.

The differences between the evaluation of Manninen et al. 2021, and our Fig. 6D and Fig S6D is discussed starting at L476. Also, the good performance of the SAR data as evaluated by Manninen et al. 2021 is presented at L489.

Still, SAR-based estimates are sufficient to classify the landscape between wet locations impacted by the lateral groundwater flow, and drier locations (see these two groups in Fig. 6D). Our aim was not to obtain a perfect match with observed soil moisture, but to reveal where lateral flow and groundwater dynamics are potentially impacting the surface soil moisture dynamics. If we compared the model only to those few in-situ point measurements, as is usually done, we would completely miss one of the main findings: most of the SpaFHy-2D modeled saturated/wet areas match the SAR estimates. Hence, they are likely saturated (or close to) in reality as well.

Please see the revised Sect. 3.2., 3.4., and 4.2. for clarified discussion on the strengths and limitations of the SAR data, and the aims of this study.

L402: Does this statement about differences being highest in wet conditions hold if the panels for homog.canopy at q = 0.5, 0.9 are blank, and negligible as stated at L409?

We meant for the difference between 2D and 1D (influence of lateral groundwater flow), revised as: "As expected, the difference between SpaFHy-2D and 1D simulations is highest in wet conditions."

Discussion

L414-416: It has been shown that the model parameterization shapes this (with improved but not strong performance in 2D), but observational data as shown/analyzed do not demonstrate this directly.

We argue that the observational data does demonstrate the impact of groundwater on soil moisture spatial and temporal dynamics, e.g. in Fig. 5 (i7, i8, i17), Fig. 7 and Fig. 8. As an example, let's consider only peat soils of which field capacity is

approximately 0.53 m$^3$m$^-$ and porosity is 0.89 m$^3$m$^{-3}$. When the measured or SAR-based estimated soil moisture content falls clearly above field capacity and close to the porosity, it is due to the impact of groundwater (or in more rare cases after heavy precipitation event that didn't yet infiltrate deeper into the soil). It is also worth noting that SAR rarely predicts full saturation as the prediction integrates multiple signals within the 16 x 16 m$^2$ grid-cell (including dry patches influenced e.g. by peatland microtopography) (mentioned at L334). Hence, majority of the measurements and SAR-based estimates that are close to porosity are influenced by groundwater dynamics.

L417: It remains hard to see from the model performance illustrated that the models are 'reliably' predicting soil moisture. That they predict moisture variability is besides the point if the predictions are not also accurate.

We understand that using 'reliably' is not ideal here, we will remove it from the sentence. While these predictions are not the the ground-truth, they are clearly less biased compared to 1D and TOP approaches (MBE: -0.09 vs. MBE: 0.03).

L425/26: The figure cited to support this statement (Figure S7) doesn't show a comparison of the 2D model with HydroGeoSphere.

In fact, we have cited both our Fig. S7 as well as Autio et al. 2023: Fig. 6 (HydroGeoSphere) here so that the reader can have a qualitative look of the simulated patterns by both approaches. We have now added the performance metrics table (see our previous reply) that can also be compared with Autio et al. 2023 Table S6.

L434: See earlier recommendation to consider these classes instead of vegetation.

See our earlier reply regarding site classes and our alternative version of Fig. 6.

L437: shallow soil moisture

Will be corrected.

L487: Yes, this is important as snowmelt is radiation rather than temperature driven, so this suggests that the process might be arriving at the right answer for the wrong reason.

You're right, and that is why we have recognized this as a limitation here: "The snowpack representation of SpaFHy successfully captured the snowmelt timing (Fig. S1, S4), but relies on a simple degree-day approach, potentially limiting its ability to fully capture snowmelt dynamics.". In fact, we are planning to include option for energy balance-based snowmelt in the next SpaFHy version.

Conclusions

L515: "shaping model simulations of soil moisture dynamics"

We are not sure if we understand this comment correctly. If the reviewer would like to specify that only the model simulation seems to be shaped by the lateral groundwater flow. We do not agree with this, as already explained in our earlier replies. The impact of groundwater dynamics is additionally demonstrated by the in-situ measurements and SAR-based data.

Given the unreliability of the SAR observations, it would be beneficial to touch on the large errors in soil moisture simulation in this section, and focus less on the SAR data. Certainly, the model progression has led to the ability to simulate a wider range of moisture conditions, but given the performance demonstrated, the model predictions are probably not robust enough to see applied use yet. Given this, a strong argument should be made in the conclusion for continued model performance to improve on that shown here.

As explained earlier, we argue that the SAR-based data is sufficient for the use it was intended in this study. We do however want to mention its limitations in the conclusion for further improving spatial soil moisture monitoring.

We are going to add at L518: "However, even the improved soil moisture simulations were affected by uncertainties in hydraulic parameters, which were estimated based on geospatial data on soil types."

See L546 in the revised manuscript.

Figures and Tables

Figure 1. "and its hydrological measurement stations"

We prefer to revise as 'hydrological measurement locations' as the soil moisture measurement locations are not fixed stations.

Tables in supplementary information should be labelled with S, to distinguish from the manuscript.

Will be corrected.

Table S1 and S2 should read "Soil type–specific"

Thank you for pointing this out.

Table 2: This is not a complete list. At a minimum a more descriptive caption is needed here that directs the reader to additional model parameters provided in the SI.

Citation to the supplementary tables will be added.

Figure 3. These panels are too small to be legible at print scale. It seems that panel B and C should be presented first, as this summarizes raw data, while the other two panels are results oriented. What period is captured by panels B and C? On panel A, why is snow presented as a line, rather than having Psnow and Pliquid as stacked bar plot to give total P. This doesn't allow for easy interpretation. Are snowpack observations available to evaluate model predictions?

Text size in all the panels will be increased for easier readability and the order of the panels will be changed as suggested by the reviewer:

[Figure]

*Figure 2. Hydrometeorological characteristics of Pallas. (A) Monthly observed climatology for the simulation period, (B) monthly observed volumetric soil moisture ($\theta$) and snow depth (HS) for the simulation period at Kenttärova forest site. The air temperature (T) and soil moisture envelopes represent minimum and maximum monthly averages of different years, while the snow depth envelope shows minimum and maximum of monthly maximums of different years. (C) Annual water budget as observed (obs) and simulated (mod) with SpaFHy-2D, where $Q_{obs}$ is observed runoff, $ET_{mod}$ is simulated evapotranspiration, $P_{liquid,obs}$ is observed precipitation, and $P_{snow,mod}$ is modelled solid precipitation. The change in catchment water storage (including canopy water, soil water and groundwater storage) dS/dt = P + ET + Q is not shown. Due to gaps in runoff measurements in 2018, runoff observation is not presented.*

Period presented in B and C is the simulation period (same as in A). We will add this in the caption.

Precipitation bar plot shows all the measured precipitation (regardless of the phase). The amount of simulated solid precipitation is then presented as a line. If we were to illustrate these as stacked plots, then the $P_{obs}$ wouldn't show the actual total observed precipitation but instead $P_{obs} – P_{snow,mod}$. Hence, we prefer to keep it this

way. However, thank you for pointing this out, as we noticed that the legend $P_{liquid,obs}$, should be $P_{obs}$.

Please find the revised Fig. 3 and caption in the revised manuscript.

Snow water equivalent observations at Kenttärova and Lompolojänkkä are evaluated in Fig. S4.

Figure 7. While described as qualitative, this figure isn't particularly easy to interpret, as it isn't always easy to distinguish between points and the underlying land use. As this largely conveys the same information as Figure 6, but less effectively, this seems a good candidate to move to the Supporting information.

We understand the difficulty to distinguish between the land-use, and for that an additional comparison to Fig. 2 can be helpful, as cited in the manuscript. The purpose of this figure is to address spatial soil moisture variability between the model versions and against the spatially distributed measurements. We believe this supports the findings in Fig. 6, but also adds value in representing the full soil moisture variability as simulated (not only those measurement locations).

Figure 8: "Spatial patterns". This plot is hard to evaluate. Please include performance metrics to allow diagnosis of model performance.

Thank you, the typo will be corrected. Note that the performance metrics of all manual in-situ measurements against model versions and SAR are provided in Fig. 6. We will also add the version of Fig. 6 with soil type classification to the supplement (see earlier reply with the new figure). In addition, we have made density scatterplots to further compare SpaFHy-2D and SAR (separated for peat and mineral soils), including mean absolute difference (MAD) and mean difference (MD) metrics (see below). This figure will be included in the supplement of the revised manuscript. As can be noted, SpaFHy-2D and SAR agree relatively well on most points on the peat soils. In addition, this further confirms that both SAR and SpaFHy are divided between two groups on peatlands; cluster of wet points impacted by the groundwater dynamics and cluster of drier points not impacted by the lateral water flow. This will be mentioned at L394: "This is also supported by a quantitative comparison of SpaFHy-2D and SAR estimates in Fig. SX; both SAR and SpaFHy-2D are divided between two groups on peatlands; cluster of wet points impacted by the lateral groundwater dynamics and cluster of drier points not impacted by the lateral flow."

However, when comparing the soil moisture estimates based on SAR to those based on the model, one must consider that the SAR based soil moisture value of a model pixel is an average of all values of original SAR pixels within the model pixel, whereas the model considers the pixel homogeneously with one soil moisture

value. Inevitably averaging the original SAR based soil moisture estimates reduces the variation range of the soil moisture values. For that reason, the large soil moisture estimates based on SAR tend to be smaller than those of the model.

[Figure]

*Figure 3. Density scatterplots of SpaFHy-2D vs. SAR on mineral (left column) and peat (right column) soil on 2019-06-26 (first row) and 2019-08-01 (second row). Mean absolute difference (MAD) and mean difference(MD) are presented in each panel.*

See Fig. S8 and text starting at L408 in the revised manuscript.

Figure 10. An improved caption is needed here. This isn't demonstrating lateral groundwater flow, but rather model parameterization that includes this process. It isn't immediately clear why panels E and F are blank.

We will revise that these are demonstrated by simulations, and mention that the blank means 0.0: "Figure 10. The impact of lateral groundwater flow (upper row) on rootzone soil moisture expressed as simulated $\Delta\theta$ = 2D - 1D, and the impact of vegetation heterogeneity (bottom row) expressed as simulated $\Delta\theta$ = 1D -

$1D_{homog.canopy}$ in different catchment soil moisture states. The panels correspond to 0.1, 0.5, and 0.9 quantiles of grid-cell soil moisture, and the bars show distribution of binned differences. Mean difference (MD) is shown in each panel. Note that the blank refers to $\Delta\theta = 0$ $m^3m^{-3}$."

General comments

There are places where hyphens are used where negative symbols are needed.

Thank you for pointing this out, these will be double-checked.

Notation style should be harmonized, e.g. there are instances of unit/unit$^2$ and unit unit$^{-2}$

These will be harmonized in the revised manuscript.

It would be helpful to have the Figures in the SI appear in the same order in which they are cited in the text.

Indeed, the order of supplement materials will be revised.

References

Ala-aho, P., Tetzlaff, D., McNamara, J. P., Laudon, H., and Soulsby, C.: Using isotopes to constrain water flux and age estimates in snow-influenced catchments using the STARR (Spatially distributed Tracer-Aided Rainfall–Runoff) model, Hydrol. Earth Syst. Sci., 21, 5089–5110, https://doi.org/10.5194/hess-21-5089-2017, 2017.

Shellito, P. J., and Coauthors, 2020: Assessing the Impact of Soil Layer Depth Specification on the Observability of Modeled Soil Moisture and Brightness Temperature. J. Hydrometeor., 21, 2041–2060, https://doi.org/10.1175/JHM-D-19-0280.1.

Tyystjärvi, V., Kemppinen, J., Luoto, M., Aalto, T., Markkanen, T., Launiainen, S., Kieloaho, A.-J., & Aalto, J. (2022). Modelling spatio-temporal soil moisture dynamics in mountain tundra. Hydrological Processes, 36(1), e14450. https://doi.org/10.1002/hyp.14450

Launiainen, S., Guan, M., Salmivaara, A., and Kieloaho, A.-J.: Modeling boreal forest evapotranspiration and water balance at stand and catchment scales: a spatial approach, Hydrol. Earth Syst. Sci., 23, 3457–3480, https://doi.org/10.5194/hess-23-3457-2019, 2019

Launiainen, S., Kieloaho, A. J., Lindroos, A. J., Salmivaara, A., Ilvesniemi, H., & Heiskanen, J. (2022). Water Retention Characteristics of Mineral Forest Soils in Finland: Impacts for Modeling Soil Moisture. Forests, 13(11). https://doi.org/10.3390/f13111797

Marttila, H., Lohila, A., Ala-Aho, P., Noor, K., Welker, J. M., Croghan, D., Mustonen, K., Meriö, L., Autio, A., Muhic, F., Bailey, H., Aurela, M., Vuorenmaa, J., Penttilä, T., Hyöky, V., Klein, E., Kuzmin, A., Korpelainen, P., Kumpula, T., ... Kløve, B. (2021). Subarctic catchment water storage and carbon cycling – Leading the way for future studies using integrated datasets at Pallas, Finland. Hydrological Processes, 35(9), 1–19. https://doi.org/10.1002/hyp.14350

Manninen, T., Jaaskelainen, E., Lohila, A., Korkiakoski, M., Rasanen, A., Virtanen, T., Muhic, F., Marttila, H., Ala-Aho, P., Markovaara-Koivisto, M., Liwata-Kenttala, P., Sutinen, R., & Hanninen, P. (2021). Very High Spatial Resolution Soil Moisture Observation of Heterogeneous Subarctic Catchment Using Nonlocal Averaging and Multitemporal SAR Data. IEEE Transactions on Geoscience and Remote Sensing, 1–17. https://doi.org/10.1109/TGRS.2021.3109695

Knoben, W. J. M., Freer, J. E., and Woods, R. A.: Technical note: Inherent benchmark or not? Comparing Nash–Sutcliffe and Kling–Gupta efficiency scores, Hydrol. Earth Syst. Sci., 23, 4323–4331, https://doi.org/10.5194/hess-23-4323-2019, 2019.

Autio, A., Ala-Aho, P., Rossi, P. M., Ronkanen, A.-K., Aurela, M., Lohila, A., Korpelainen, P., Kumpula, T., Klöve, B., & Marttila, H. (2023). Groundwater exfiltration pattern determination in the sub-arctic catchment using thermal imaging, stable water isotopes and fully-integrated groundwater-surface water modelling. Journal of Hydrology, 626, 130342. https://doi.org/https://doi.org/10.1016/j.jhydrol.2023.130342

Douinot, A., Tetzlaff, D., Maneta, M., Kuppel, S., Schulte-Bisping, H., & Soulsby, C. (2019). Ecohydrological modelling with EcH2O-iso to quantify forest and grassland effects on water partitioning and flux ages. Hydrological Processes, 33(16), 2174–2191. https://doi.org/10.1002/hyp.13480

**Author's reply to Referee #3**

This is my first review of the paper "Multi-scale soil moisture data and process-based modeling reveal the importance of lateral groundwater flow in a subarctic catchment" by Jari-Pekka Nousu et al.

The paper is well-written and presents a valuable contribution to the literature. As models and data improve in resolution, many processes become scale-dependent, making what is overlooked at a coarse scale crucial at high resolution. This manuscript addresses this issue by comparing different model parameterizations and SAR-based soil moisture with a robust experimental dataset.

I do not have major comments on the study, but I suggest some moderate revisions:

We would like to thank the reviewer for the positive assessment and comments on our study. Our answer to each comment is written in blue. Responses to the annotated manuscript is provided as answers to each comment in a separate PDF file.

Update the Bibliography: The bibliography is outdated. Please revise it to include more recent works. I have provided some suggestions in the annotated PDF.

Thank you for providing these relevant suggestions. We will update the bibliography accordingly.

Please see the revised manuscript with the updated references and bibliography. Once again, thank you for these suggestions.

Enhance Section 3.4: Section 3.4 is overly qualitative and could be improved significantly. Consider incorporating metrics to quantitatively demonstrate the differences between SAR data, various model parameterizations, and in situ data. Temporal stability analysis, as discussed in Dari et al. 2019 (https://www.sciencedirect.com/science/article/abs/pii/S0022169419300575), could be particularly useful. Comparing different statistical spatial measures from various soil moisture spatiotemporal dynamics would be highly relevant.

Thank you for your valuable insights. We agree that quantitative material will improve section 3.4. Metrics of model versions and SAR against in-situ data are already provided in Fig. 6 of the manuscript. We have made an alternative version of Fig. 6 to better quantify the performance of the model and SAR within specific soil types (see below). This new figure will be included in the supplement of the revised manuscript. This figure highlights the uncertainties in soil parameterization based on the geospatial soil type data: many of the peat soil points of the 2D approach are overestimated (few also underestimated). We will highlight the uncertainties in soil parameters in the revised manuscript.

[Figure]

*Figure 4. Comparison of simulated rootzone soil moisture content and SAR-based surface soil moisture estimates against spatiotemporal manual in-situ soil moisture observations. The blue color of the points correspond to peat soil and the orange color to mineral soil.*

Please find Fig. S6 in supplement.

The temporal stability analysis proposed by Dari et al., 2019 is indeed an interesting approach, and we appreciate the suggestion. However, there are many approaches that can be useful for our specific case. To bring in more quantitative analysis, we have calculated relevant metrics of the model predictions and SAR estimates and summarized them in the table below.

**Table 5.** Statistics of SpaFHy-1D, SpaFHy-TOP, SpaFHy-2D and SAR morning estimates. Low and high soil moisture quantiles are represented as q = 0.1 and q = 0.9, respectively. All statistics were calculated for those days when SAR morning estimates were available.

| Data | mean ($m^3 \, m^{-3}$) | variance ($m^3 \, m^{-3}$) | q = 0.1 ($m^3 \, m^{-3}$) | q = 0.9 ($m^3 \, m^{-3}$) |
|---|---|---|---|---|
| SpaFHy-1D | 0.29 | 0.01 | 0.25 | 0.47 |
| SpaFHy-TOP | 0.30 | 0.01 | 0.25 | 0.47 |
| SpaFHy-2D | 0.39 | 0.04 | 0.25 | 0.82 |
| SAR | 0.34 | 0.02 | 0.22 | 0.65 |

In addition, we have made a density scatterplot comparison of SpaFHy-2D and SAR for peat and mineral (medium texture) soils, including metrics of mean absolute difference (MAD) and mean difference (MD) (see below). This figure will be included in the supplement of the revised manuscript and referred to at L394: "This is also supported by a quantitative comparison of SpaFHy-2D and SAR estimates in Fig. SX; both SAR and SpaFHy-2D are divided between two groups on peatlands; cluster of wet points impacted by the lateral groundwater dynamics and cluster of drier points not impacted by the lateral flow."

When comparing the soil moisture estimates based on SAR to those based on the model, one must consider that the SAR based soil moisture value of a model pixel is an average of all values of original SAR pixels within the model pixel, whereas the model considers the pixel homogeneously with one soil moisture value. Inevitably averaging the original SAR based soil moisture estimates reduces the variation range of the soil moisture values. For that reason, the lowest soil moisture

estimates based on SAR tend to be larger than those of the model and the large soil moisture estimates based on SAR tend to be smaller than those of the model.

[Figure]

Figure 5. Density scatterplots of SpaFHy-2D vs. SAR on mineral (left column) and peat (right column) soil on 2019-06-26 (first row) and 2019-08-01 (second row). Mean absolute difference (MAD) and mean difference(MD) are presented in each panel.

Please see Fig. S8 and Table S5 in supplement, and L392-397 and L408-412 in the revised manuscript.

Clarify SAR Estimates: While there is already a paper on SAR estimates, more detailed information about the retrievals should be included in this manuscript to provide better context.

The SAR retrieval method was based on the gradient boosting machine learning method and the following input variables were used: 1) day/time of soil moisture to be calculated, 2) time difference between the SAR image acquisition and the time for which the soil moisture is to be calculated, 3) altitude of terrain, 4) local slope of terrain, 5) local aspect angle of terrain, 6) land cover class, 7) local incidence angle of SAR, 8) azimuth difference of SAR looking direction and terrain slope, 9) leaf area index estimate based on SAR image, 10) cosine of SAR incidence angle, 11) VH

backscattering coefficient of SAR pixel, 12) VV backscattering coefficient of SAR pixel, 13) average VH backscattering coefficient of land areas in SAR image, 14) average VV backscattering coefficient of land areas in SAR image. The leaf area index estimation algorithm for SAR is another gradient boosted algorithm that is trained with the reduced simple ratio (RSR) index based on an optical image. The land cover class is one of 30 classes derived from the multitemporal statistics of the SAR images in the area. The backscattering coefficient values of SAR used as input for the soil moisture and effective LAI retrieval methods are nonlocally averaged using the PIMSAR method (Manninen and Jääskeläinen 2021).

The description starting at L269 is slightly modified to give the previous overall information without going too much in details; "The soil moisture retrieval using SAR images is based on the gradient boosting machine learning method using as input variables nonlocally averaged VH and VV backscattering coefficients, multitemporal SAR statistics, terrain data, effective LAI estimates based on SAR, SAR overpass information and date/time for soil moisture estimate to be calculated. It is validated against discrete and continuous in-situ soil moisture measurements at Pallas (Manninen et al., 2021)."

See revised text at L282 of the revised manuscript.

Based on these points, my recommendation is moderate revisions. I have also attached the annotated PDF with additional comments for further guidance.

Thank you for the revision, and the additional comments. These comments will be considered in order to further improve the manuscript.

References

T. Manninen and E. Jääskeläinen, "Pixel Based Multitemporal Sentinel-1 SAR Despeckling PIMSAR," in IEEE Geoscience and Remote Sensing Letters, vol. 19, pp. 1-5, 2022, Art no. 4011705, doi: 10.1109/LGRS.2021.3065300. keywords: {Backscatter;Indexes;Synthetic aperture radar;Standards;Radar polarimetry;Spatial resolution;Wetlands;Land surface;synthetic aperture radar (SAR) data;vegetation},

Dari, J., Morbidelli, R., Saltalippi, C., Massari, C., & Brocca, L. (2019). Spatial-temporal variability of soil moisture: Addressing the monitoring at the catchment scale. Journal of Hydrology, 570, 436–444. https://doi.org/10.1016/J.JHYDROL.2019.01.014